# A high-resolution mRNA expression time course of embryonic development in zebrafish

Richard J White[1], John E Collins[1], Ian M Sealy[1], Neha Wali[1], Christopher M Dooley[1], Zsofia Digby[1†], Derek L Stemple[1], Daniel N Murphy[2], Konstantinos Billis[2], Thibaut Hourlier[2], Anja Füllgrabe[2], Matthew P Davis[2], Anton J Enright[2‡], Elisabeth M Busch-Nentwich[1,3]*

[1]Wellcome Trust Sanger Institute, Hinxton, United Kingdom; [2]European Molecular Biology Laboratory, European Bioinformatics Institute, Hinxton, United Kingdom; [3]Department of Medicine, University of Cambridge, Cambridge, United Kingdom

**Abstract** We have produced an mRNA expression time course of zebrafish development across 18 time points from 1 cell to 5 days post-fertilisation sampling individual and pools of embryos. Using poly(A) pulldown stranded RNA-seq and a 3′ end transcript counting method we characterise temporal expression profiles of 23,642 genes. We identify temporal and functional transcript co-variance that associates 5024 unnamed genes with distinct developmental time points. Specifically, a class of over 100 previously uncharacterised zinc finger domain containing genes, located on the long arm of chromosome 4, is expressed in a sharp peak during zygotic genome activation. In addition, the data reveal new genes and transcripts, differential use of exons and previously unidentified 3′ ends across development, new primary microRNAs and temporal divergence of gene paralogues generated in the teleost genome duplication. To make this dataset a useful baseline reference, the data can be browsed and downloaded at Expression Atlas and Ensembl.
DOI: https://doi.org/10.7554/eLife.30860.001

*For correspondence:
emb@sanger.ac.uk

Present address: †Department of Veterinary Medicine, University of Cambridge, Cambridge, United Kingdom; ‡Department of Pathology, University of Cambridge, Cambridge, United Kingdom

Competing interests: The authors declare that no competing interests exist.

## Introduction

Gene regulatory interactions are the fundamental basis of embryonic development and transcription is one of the major processes by which these interactions are mediated. A time-resolved comprehensive analysis of relative mRNA expression levels is an important step towards understanding the regulatory processes governing embryonic development. Here we present a systematic assessment of the temporal transcriptional events during this critical period in zebrafish (*Danio rerio*).

The zebrafish is a unique vertebrate model system as it possesses high morphological and genomic conservation with humans, but also experimental tractability of embryogenesis otherwise only found in invertebrate model organisms such as *Drosophila melanogaster* or *Caenorhabditis elegans*. Features such as very large numbers of offspring, ex vivo development and embryonic translucency have enabled comprehensive forward and reverse genetic screens (*Amsterdam et al., 1999*; *Driever et al., 1996*; *Haffter et al., 1996*; *Kettleborough et al., 2013*; *Moens et al., 2008*; *Varshney et al., 2015*) as well as high-throughput drug discovery approaches (*Murphey et al., 2006*; *North et al., 2007*; *Peterson et al., 2000*; *Peterson et al., 2004*; *Stern et al., 2005*). Together with a high quality genome (*Howe et al., 2013b*) only comparable in vertebrates to mouse and human, this has led to many important discoveries in areas such as zygotic genome activation (ZGA; *Lee et al., 2013*), blood stem cell biology (*Bertrand et al., 2010*; *Kissa and Herbomel, 2010*), and findings directly affecting human health (*Li et al., 2015*; *Tobin et al., 2010*).

The morphological processes underlying the transformation of a fertilised egg into a free swimming fish have been studied extensively owing to the ease with which embryogenesis can be observed and manipulated (*Behrndt et al., 2012*). This has identified many genes that drive crucial steps of the differentiation process, however the wealth of morphological phenotype data has not been matched with a systematic analysis of the corresponding molecular events. Baseline transcriptomic datasets in other species have greatly improved knowledge of relative levels of gene expression as well as alternative splicing events (*Boeck et al., 2016*; *Brown et al., 2014*; *Gerstein et al., 2014*; *Graveley et al., 2011*; *Hashimshony et al., 2015*; *Klepikova et al., 2016*; *Owens et al., 2016*; *Tan et al., 2013*).

The first baseline expression study in zebrafish was conducted by *Mathavan et al. (2005)* using microarrays. This profiled the expression of 14,904 genes across 12 time points from unfertilised egg to 2 days post-fertilisation (dpf). Other baseline transcriptome work in zebrafish has focused on either certain aspects of development such as the maternal-zygotic transition (*Aanes et al., 2011*; *Harvey et al., 2013*; *Lee et al., 2013*), the identification of specific transcript types (*Pauli et al., 2012*) or promoters (*Gehrig et al., 2009*; *Nepal et al., 2013*), on gene annotation improvement (*Collins et al., 2012*) or have a limited number of time points and replicates (*Yang et al., 2013*).

Using poly(A) pulldown RNA-seq we have generated a comprehensive polyadenylated RNA expression profile of normal zebrafish development. We have also used a 3′ end enrichment method called DeTCT, developed by us for cost-effective high-throughput polyA transcript screening (*Collins et al., 2015*), to catalogue 3′ end usage. We have taken advantage of the large numbers of offspring and have divided the 5 day period of embryonic development into 18 stages. For each stage, we have processed up to 29 biological replicates consisting of single or pooled embryos.

We have placed particular emphasis on making the presented data accessible without the need for computational processing. The data are available to browse and download in a user-friendly interface at Expression Atlas and as selectable tracks in the Ensembl genome browser. Furthermore, incorporation into the Ensembl release 90 (August 2017) genebuild has identified novel protein-coding genes, temporal profiles of ~2000 new lincRNAs and stage-specific transcript isoform annotation.

This developmental polyadenylated RNA baseline reference enhances our understanding of the gene regulatory events during vertebrate development and provides a time-resolved transcriptional basis for the comprehensive annotation of functional elements in the zebrafish genome using existing and future genome-wide datasets (*Haberle et al., 2014*; *Kaaij et al., 2016*; *Lindeman et al., 2010*; *Potok et al., 2013*; *Vastenhouw et al., 2010*).

## Results

### A high-resolution transcriptional profile of zebrafish development

We sampled a total of 18 developmental time points in this study (*Figure 1A*). Four time points are before or at the onset of zygotic transcription, four during gastrulation, three during somitogenesis, three during prim stages – 24, 30 and 36 hr post-fertilisation (hpf) – and every 24 hr from 2 to 5 dpf. This gives detailed coverage of all the important developmental processes taking place during this time period. At each time point, five biological replicates of pools of 12 embryos were collected and we obtained an average of 3.8 million reads per sample (*Figure 1—figure supplement 1A*). The number of genes detected rises steadily as development proceeds and a wider range of genes are expressed (*Figure 1—figure supplement 1B*). The reduced number of detected genes during gastrulation is most likely caused by degradation of maternal mRNAs. The distribution of expression levels (TPM, transcripts per million) is initially bimodal, with one peak at very low levels and another at much higher levels, and becomes more normally distributed as the embryo develops (*Figure 1—figure supplement 1C*, *Supplementary files 2–3*). This pattern is consistent with the observation that RNA-seq on homogeneous populations of cells produces a bimodal distribution whereas the distribution becomes more unimodal when heterogeneous samples are sequenced (*Hebenstreit et al., 2011*), which fits well with the cells of the embryo being very similar at early stages and becoming more differentiated over time.

A sample correlation matrix produced by comparing expression levels across all genes between samples demonstrates the excellent agreement among biological replicates and close relation of

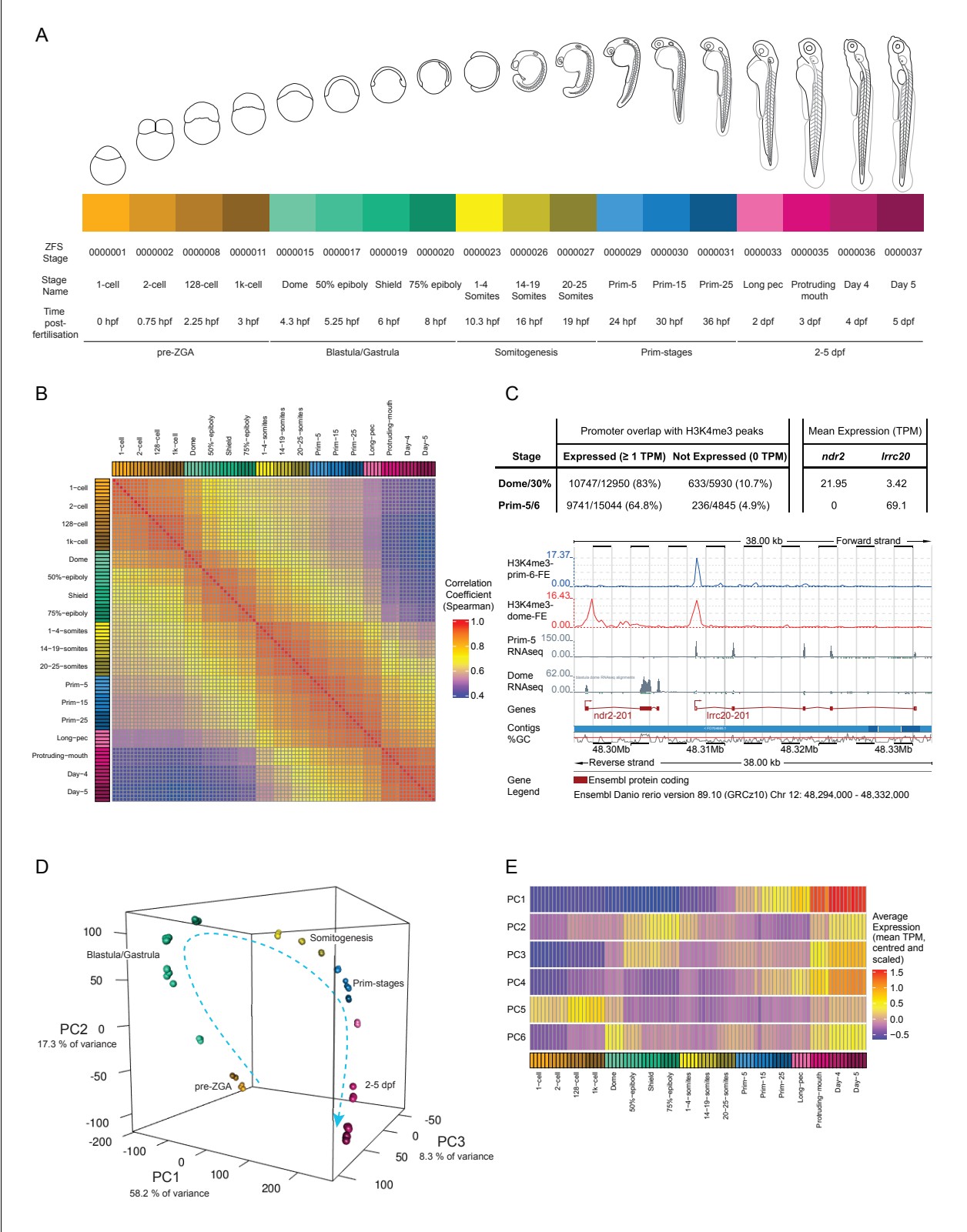

**Figure 1.** A transcriptional map of development. (**A**) Stages represented in this study. The colour scheme is used throughout the figures. Also shown are the ZFS stage IDs, stage names and approximate hours post-fertilisation (at 28.5°C) for each stage and five developmental categories. (**B**) Sample correlation matrix of Spearman correlation coefficients of every sample compared to every other sample. (**C**) Example of the overlap between H3K4me3 signal and genes. Ensembl screenshot of the region occupied by *ndr2* and *lrcc20* (the direction of transcription is indicated by arrows). Included tracks

*Figure 1 continued on next page*

*Figure 1 continued*

are: H3K4me3-prim-6-FE (blue) – fold enrichment of H3K4me3 ChIP-seq (H3K4me3/control) at prim-6 stage, H3K4me3-dome-FE (red) – fold enrichment of H3K4me3 ChIP-seq (H3K4me3/control) at dome/30% epiboly, Prim-5 RNAseq and Dome RNAseq (grey) – alignments of RNA-seq reads. The table above shows the genome-wide overlap between H3K4me3 ChIP-seq peaks and the promoters of expressed/not expressed genes as well as the average expression (TPM) of each gene at each stage. *ndr2* is expressed only at dome stage and displays a peak only in the dome/30% epiboly H3K4me3 ChIP-seq. *lccr20* is expressed at both stages and has a H3K4me3 peak at both stages. (D) 3D plot of Principal Component Analysis (PCA) showing the first three principal components (PCs), which together explain 83.8% of the variance in the data. Samples from the same stage cluster together and there is a smooth progression through developmental time (shown as dashed arrow). Samples are coloured as in (A) and are annotated with the stage categories. The amount of variance explained by each PC is indicated on each axis. (E) Representation of the expression profiles that contribute most to the first 6 PCs. The expression values are centred and scaled for each gene for the 100 genes that contribute most to that PC and then the mean value for each sample is plotted.

DOI: https://doi.org/10.7554/eLife.30860.002

The following figure supplements are available for figure 1:

**Figure supplement 1.** RNA-seq sample QC.
DOI: https://doi.org/10.7554/eLife.30860.003
**Figure supplement 2.** PCA matrix plot.
DOI: https://doi.org/10.7554/eLife.30860.004

neighbouring stages in terms of gene expression profiles (*Figure 1B*). As a further validation of the dataset we overlaid published Chromatin ImmunoPrecipitation sequencing (ChIP-seq) data on H3K4me3 status (*Haberle et al., 2014*). Of the 12,950 genes detected at 1 TPM or above at dome, 10,747 (83.0%) of the promoter regions (1000 bp upstream to 200 bp downstream of the annotated transcription start site) overlap with an H3K4me3 peak, whereas only 10.7% (633/5930) of the promoters of genes that are not expressed (0 TPM in all samples at that stage) overlap (*Figure 1C*). Similarly, at prim-5 (24 hpf) 64.8% (9741/15,044) of expressed genes have a corresponding peak, but only 4.9% (236/4845) of genes that are not expressed. The overlap of expressed genes at this later stage is lower than at dome. However, this is probably due to the higher differentiation and therefore tissue-restricted gene expression in embryos at prim-5. Whole embryo H3K4me3 ChIP-seq is not as sensitive as RNA-seq in detecting spatially restricted gene expression as it interrogates a small stretch of genomic sequence that carries the mark in only a subset of cells. By contrast, RNA-seq assays multiple copies of full length mRNA per gene.

Principal component analysis (PCA) shows close clustering of biological replicates and a smooth transition from one stage to another through developmental time (*Figure 1D*). The first six principal components (PCs) explain over 90% of the variance in the data (*Figure 1—figure supplement 2*); the first component accounts for 58.2% of the observed variation. Summaries of the expression patterns underlying the PCs are displayed in *Figure 1E*. Broadly, the genes that contribute the most to the first PC are either low at early time points and increase towards the end of the time course or the inverse (high early and decreasing later). Genes contributing to PC2 and PC3 have two peaks of expression at similar stages (blastula/gastrula and days 3–5) although the patterns are different. The other patterns are more complex (*Figure 1—figure supplement 2*) although PC6 has a clear spike of expression, which starts at the 1000 cell stage (3 hpf) and peaks at dome (4.3 hpf) coinciding with the start of zygotic transcription.

In order for this dataset to be as useful as possible to the scientific community, we have made it available as sequence with metadata through the European Nucleotide Archive and in multiple locations for viewing. Pre-computed count profiles (TPM or FPKM) are available to search/browse at Expression Atlas (http://www.ebi.ac.uk/gxa/experiments/E-ERAD-475)(*Petryszak et al., 2016*), allowing comparison of the expression profiles of multiple genes across the entire time course (*Figure 2A*). This is searchable by gene/allele name, Ensembl ID and Gene Ontology (GO) terms. It is also possible to start with a single seed gene and have Expression Atlas add in similarly expressed genes (as assessed by k-means clustering) for comparison (https://www.ebi.ac.uk/gxa/FAQ.html#similarExpression). The data are also viewable in Ensembl as separate stage-specific selectable tracks along with one merged track (http://www.ensembl.org/Danio_rerio). The aligned reads can be displayed as either coverage graphs or read pairs. Reads that span introns can be viewed to investigate alternative splicing in a stage-specific manner (*Figure 2B–C*).

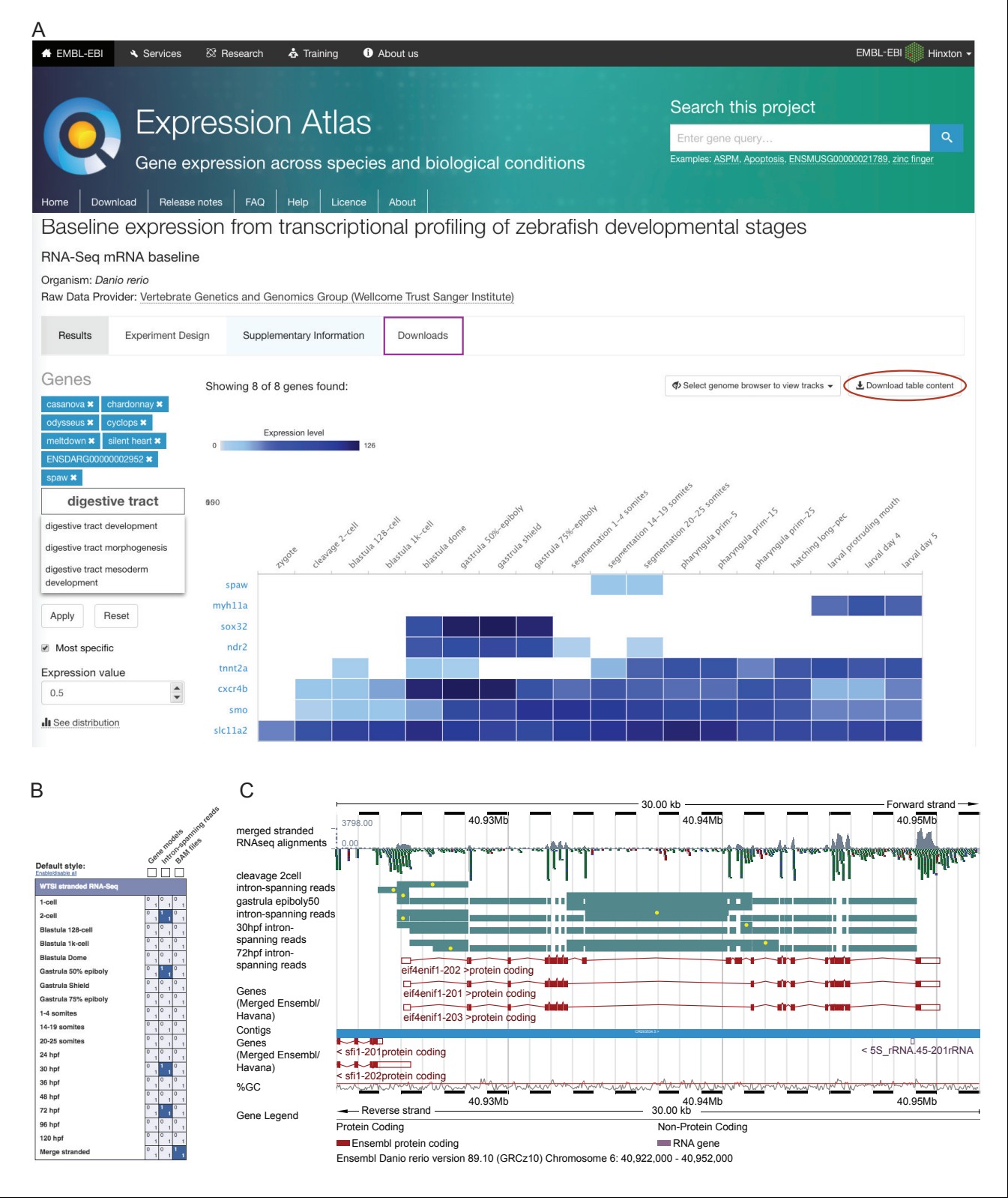

**Figure 2.** Data access. (**A**) A screenshot from Expression Atlas release May 2017. Relative gene expression levels (FPKM) across all 18 stages are displayed as a heatmap. Genes can be searched for using Ensembl IDs, gene/allele names and Gene Ontology terms. The data for the selected genes can be downloaded (red ellipse) and the entire dataset can be downloaded from the 'Downloads' tab (purple rectangle) as either a .tsv or .Rdata file. (**B**) A screenshot of the RNA-seq matrix (Ensembl v89) with four intron tracks plus the exon read coverage from the merged (all 18 stages) RNA-seq data

*Figure 2 continued on next page*

*Figure 2 continued*

chosen for display. (**C**) A screenshot of the Ensembl v89 browser showing the region surrounding the gene *eif4enif1* (chr 6:40.922–40.952 Mb). Exon read coverage from the merged 18 stages is displayed as grey histograms. Four of the 18 intron-defining tracks from the RNA-seq matrix are selected. Introns are coloured in teal. Yellow dots indicate introns identifying a splice variant not found in any of the three gene models shown in red. Note that only the forward strand is shown in full and a gene model overlaps on the reverse strand.

DOI: https://doi.org/10.7554/eLife.30860.005

## Graph-based clustering produces compact clusters of highly biologically related genes

To visualise the expression of all genes across development we ordered the genes in our dataset by their stage of maximum expression (*Figure 3A*). This organises the genes temporally, broadly by expression pattern.

Of the 23,642 detectable genes in this time course (>0 TPM in all samples for at least one stage), 5024 are genes with automatically generated names (often derived from clone IDs) such as zgc:* and si:ch*. Interestingly, at dome and gastrula stages, the proportion of these unnamed genes is nearly twice as high (32.9–42.2%) as at other stages (18.2–28.6%; *Figure 3B*). This could indicate a lack of orthologues and therefore that zebrafish gastrulation is enriched for the expression of genes exclusive to this species. However, we did not find evidence of an increased proportion of danio/teleost-specific genes at these stages (*Figure 1—figure supplement 1*). While the maximum expression stage groupings provide a high-level, temporally organised view of development, smaller more specific groupings provide more insight especially into the role of uncharacterised genes. To get smaller clusters that are more likely to contain biologically related genes, we used the BioLayout *Express*[3D] software (http://www.biolayout.org/) (*Theocharidis et al., 2009*) to cluster and visualise the dataset (*Figure 3C*). The software first builds a network graph in which genes are nodes with edges between those whose expression is correlated above a given threshold. It then uses the Markov Cluster Algorithm (MCL, *Enright et al., 2002*; *van Dongen, 2000*) to find strongly connected components within the graph (*Supplementary files 4–5*). These are clusters of genes that are more highly connected with each other than with the rest of the graph. This produces 254 clusters, the vast majority of which (238/254) have fewer than 100 genes (*Figure 3B*). Broadly speaking, the clusters on the left-hand side of the graph represent genes expressed at early stages, whereas towards the right of the graph the expression profiles are progressively later in development (*Figure 3B*). For example, cluster 3 contains genes that are maternally supplied and then degraded and cluster 5 genes are not expressed until the last time points in the study. Cluster 2 has genes that accumulate after the 2 cell stage, but before the onset of zygotic transcription (*Figure 3C*). Genes whose abundance changes before ZGA have been identified before (*Mathavan et al., 2005*) and were shown to be regulated by the control of polyadenylation (*Aanes et al., 2011*; *Harvey et al., 2013*).

The BioLayout clusters were analysed for enrichment of GO and Zebrafish Anatomy Ontology (ZFA) terms. 70 of the clusters have significant GO enrichments (adjusted p-value<0.05) and 17 have significant ZFA enrichments. For example, cluster 27 contains a large number of homeobox transcription factors (including *fox*, *hox*, *irx* and *nkx* gene families) that are expressed from 1 to 4 somites onwards and is associated with GO terms for transcription and development (*Figure 3D*). Clusters 36 and 49 have two peaks of expression and contain complement, clotting factor genes and serpins. Both clusters have enrichments for the terms related to blood clotting (*Figure 3D*). Notably, there are clusters that have stage-specific expression profiles, but no GO or ZFA enrichments (e.g. Cluster 30, *Figure 3D*). This cluster contains well known developmental regulators such as *admp*, *chd* and *fzd8b* and a large proportion of unnamed genes (10/42). The information on GO/ZFA enrichments for every cluster is provided in *Supplementary file 6*, which can be viewed in any web browser.

GO analysis on the genes which are not present in the heatmap in *Figure 3A* (0 TPM in at least one sample for all stages) produces enrichments for terms related to apoptosis and the adaptive immune system with B-cell related terms such as B cell receptor signalling, immunoglobulin production and antigen processing and presentation (*Supplementary file 7*). These fit well with the observation that the adaptive immune system does not fully develop in zebrafish until 3 weeks post-fertilisation (*Lam et al., 2004*; *Trede et al., 2004*; *Willett et al., 1997*). Although it is possible that

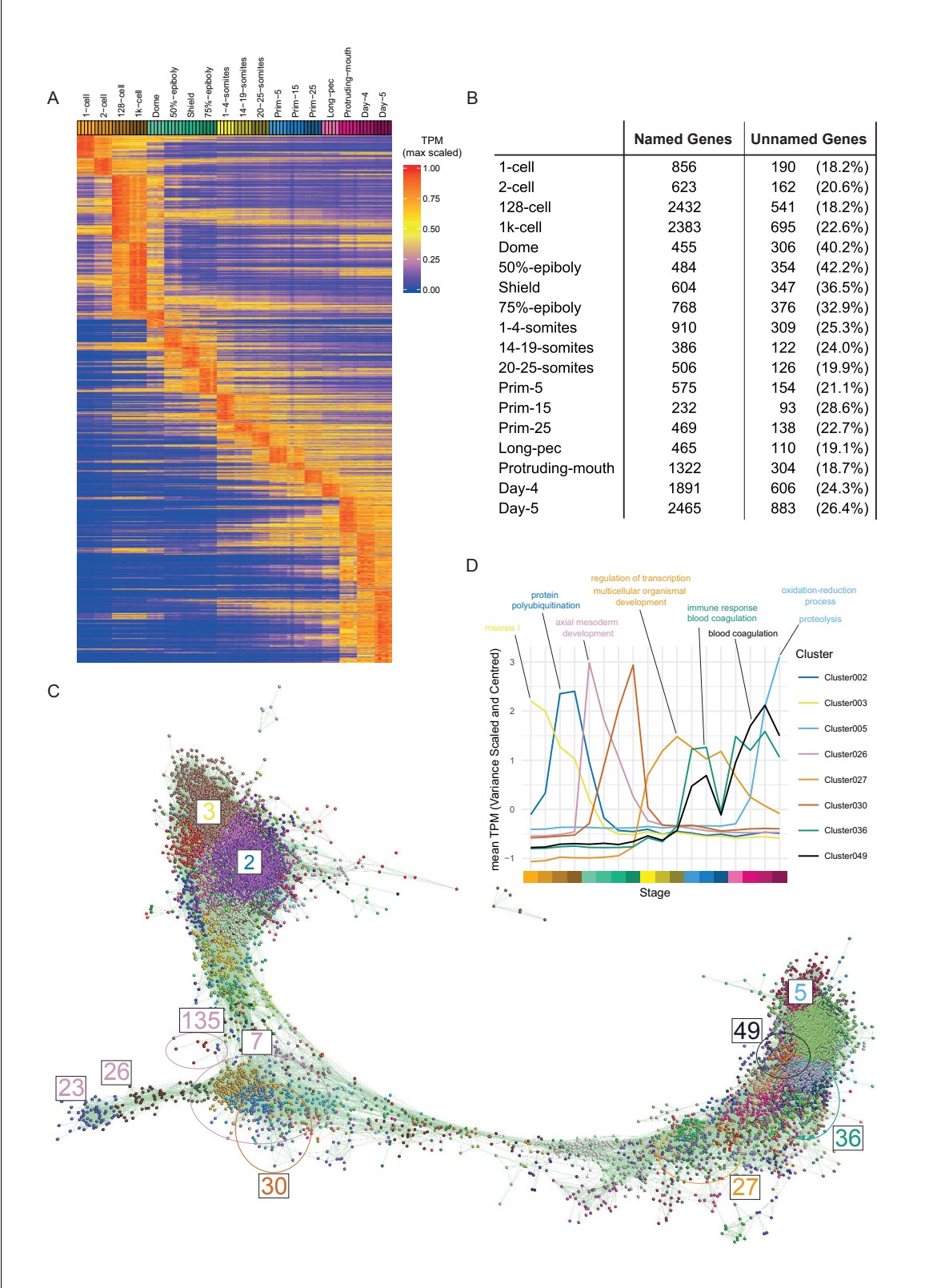

**Figure 3.** Clustering of expression patterns. (**A**) Heatmap of the expression profiles across the time course (includes genes expressed at >0 TPM in all the samples for at least one stage). The expression values are scaled to the maximum expression for that gene across the time course. Genes are organised by which stage the maximum expression occurs at and clustered within each stage. (**B**) Table of the numbers of named and unnamed genes assigned to each stage. (**C**) Network diagram of the clustering produced by MCL on a network graph generated by linking genes with a Pearson

*Figure 3 continued on next page*

*Figure 3 continued*

correlation coefficient of >0.94 and removing unassigned genes and clusters with five genes or fewer. The numbers indicate the positions in the network diagram of the clusters shown in D and *Figure 4*. (D) Example clusters illustrating the progression of expression peaks through development displayed as the mean of mean-centred and variance-scaled TPM for the genes in the clusters.

DOI: https://doi.org/10.7554/eLife.30860.006

The following figure supplement is available for figure 3:

**Figure supplement 1.** Relationship of named and unnamed genes to other species.

DOI: https://doi.org/10.7554/eLife.30860.007

this represents lack of detection rather than true absence of expression we do detect other genes expressed in macrophages (another similarly small population of cells) such as *marco* and *mpeg1.2*.

## A burst of transcription of highly related zinc finger proteins during zygotic genome activation

During examination of the clusters produced by BioLayout, we noticed several clusters protruding from the rest of the network (*Figure 3B*; clusters 23 and 26 [bottom left]) that appeared to be enriched for a family of zinc finger (ZnF) proteins located on chromosome 4 (*Howe et al., 2016*). To test systematically if any chromosomes were over-represented in any of the clusters, we performed a binomial test for each chromosome for each cluster that contained more than five genes from that chromosome. This identifies four clusters (clusters 7, 23, 26 and 135) with more chromosome 4 genes than would be expected by chance given the size of the clusters (*Table 1*). Most of the chromosome 4 genes (153/168) within these clusters are predicted to encode proteins with ZnF domains (*Supplementary file 8*). The unnamed genes within these clusters represent on average 30.6% of the unnamed genes assigned to gastrula stages in the maximum-stage groupings.

The expression profiles of these ZnF genes have a burst of expression from dome to 75% epiboly (4.3–8 hpf) and several begin to be expressed at the 128- and 1000 cell stages (*Figure 4A–C*). Cluster 23 has a particularly sharp peak with a very large increase from the 1000 cell stage to dome and an almost equal decrease from dome to 50% epiboly. This suggests that these ZnF genes are co-ordinately expressed at the onset of zygotic transcription. The effect can even be seen on the chromosomal scale (*Figure 4B*). To investigate whether this is a global effect on all the genes on the long arm of chromosome 4 we looked at the expression profiles of NLR family genes, another large family of proteins present on chromosome 4 (*Howe et al., 2016*). These genes are present in the same region of chromosome 4 as the ZnF genes although their distribution is slightly different, with more NLR genes towards the end of the chromosome (*Figure 4D*). Of the 311 NLR genes present in the Ensembl version 85 genebuild, 148 have expression (>0 TPM in all samples for at least one stage) during this time course and they are not expressed in the same pattern as the ZnF genes (*Figure 4E*, *Figure 4—figure supplement 1*). Only 9 of them are assigned to a BioLayout cluster.

## Chromosomal expression domains

Since the expression pattern of the chromosome 4 ZnF genes is obvious on a chromosome level plot, we wondered if it was possible to find any other chromosomal regions where the genes share a

**Table 1.** BioLayout clusters with significant chromosomal enrichments (adjusted p-value<0.05 and $\log_2$ fold enrichment [$\log_2$FE]>1).

| Cluster | Chromosome | Count | Expected | $\log_2$[FE] | Adjusted p-value |
|---------|-----------|-------|----------|----------|------------------|
| Cluster007 | Chr4 | 99 | 23.55 | 2.07 | 4.772e-35 |
| Cluster023 | Chr4 | 40 | 6.11 | 2.71 | 2.707e-23 |
| Cluster026 | Chr4 | 20 | 5.03 | 1.99 | 1.810e-06 |
| Cluster135 | Chr4 | 9 | 0.99 | 3.19 | 6.997e-07 |
| Cluster086 | Chr12 | 7 | 0.42 | 4.05 | 3.447e-06 |
| Cluster057 | Chr23 | 6 | 0.69 | 3.13 | 1.637e-03 |

DOI: https://doi.org/10.7554/eLife.30860.008

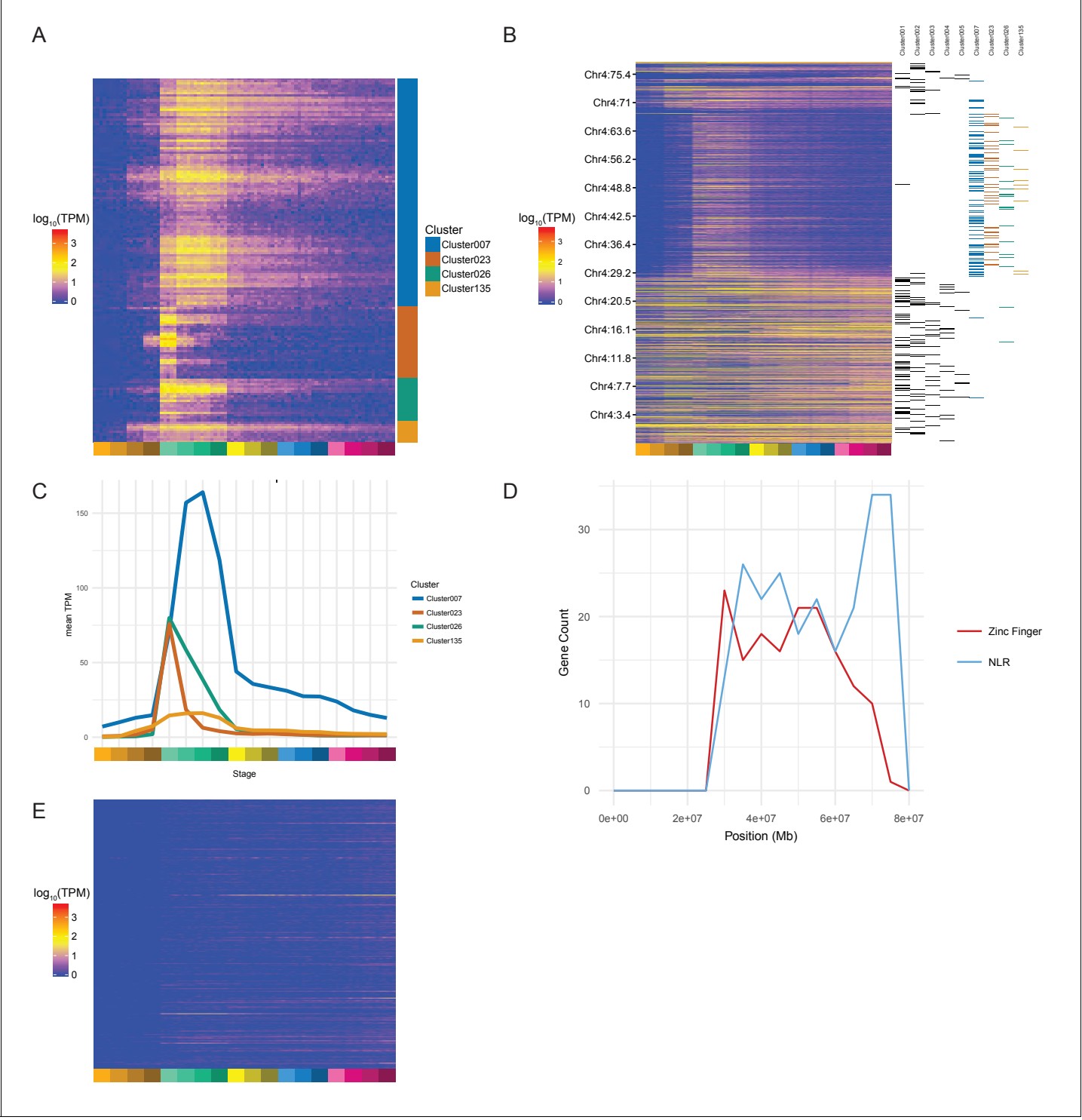

**Figure 4.** Zinc Finger (ZnF) genes expressed at the onset of zygotic transcription. (**A**) Heatmap of the expression profiles (log$_{10}$[TPM]) of the ZnF genes from clusters 7, 23, 26 and 135. (**B**) Heatmap of the expression of all genes on chromosome 4 (log$_{10}$[TPM]) in chromosomal order with cluster assignments shown on the right. (**C**) Expression profiles of clusters 7, 23, 26 and 135 shown as average expression (mean TPM). (**D**) Plot of the distribution of ZnF (red) and NLR (blue) genes on chromosome 4. (**E**) Heatmap of the expression profiles (log$_{10}$[TPM]) of the NLR genes present in GRCz10.

DOI: https://doi.org/10.7554/eLife.30860.009

The following figure supplement is available for figure 4:

**Figure supplement 1.** Comparison of ZnF and NLR genes on chromosome 4.

*Figure 4 continued on next page*

*Figure 4 continued*

DOI: https://doi.org/10.7554/eLife.30860.010

common expression pattern. To investigate this, we calculated the average pairwise Pearson correlation coefficient in a sliding window of 10 genes along each chromosome and selected regions that were above 0.5 (*Figure 5*). This identified 62 regions (*Supplementary file 9*) including the *hoxda* cluster. There are several more groups of genes containing zinc finger domains, in smaller regions on chromosomes 3, 15 and 22. These genes have a peak at a similar stage to the chromosome 4 ZnF genes, although they tend to be more widely expressed over the time course (*Figure 5—figure supplement 1A–D*).

Many of the regions contain groups of related genes that share domains such as gamma crystallins, cadherins, histone and globin genes (*Supplementary file 9*). One of the regions on chromosome 12 (*Figure 5—figure supplement 1E*) contains genes that also appear in BioLayout cluster 86, which has an overrepresentation of genes on chromosome 12 (*Table 1*). This appears to be a fish-specific expansion of cathepsin L and contains the gene *ctslb* (ENSDARG00000039173, also known as *hatching gland 1*).

For some of the regions (26/62), when compared to the mouse and human genomes, the homologous genes maintain their physical co-localisation. However most of them have been split up into two or three regions, suggesting that there is no selective pressure on maintaining synteny (*Supplementary file 9*).

## Conservation/divergence of the expression profiles of paralogous genes

Teleost fish underwent a third round of whole genome duplication compared to the two rounds of tetrapods (*Meyer and Schartl, 1999*). During rediploidisation, most of the duplicate genes were lost, but a significant proportion were retained and are now present as paralogous pairs. Paralogous genes can have conserved expression patterns but often the expression patterns have diverged so that the paralogues function at different times or in different tissues. To investigate whether we can see this in our data, which has only stage and not tissue specificity information, we produced a list of one-to-one teleost-specific paralogous pairs using data from Ensembl (2929 gene pairs with expression in our data;>0 TPM in all samples for at least one stage. *Supplementary file 10*).

We calculated the Pearson correlation coefficients between the expression values (TPM) for each paralogous pair as a measure of how similar the expression profiles are. The distribution of the correlation coefficients is skewed to high positive correlations when compared to a random sample of gene pairs of similar size (*Figure 6A*). Despite this skew, there is a significant proportion of the distribution with low or negative correlation coefficients, suggesting that this may represent a set of genes whose expression patterns have diverged (50% of the distribution is below 0.45). Examples of highly positively (*ckma/b*) and negatively (*impdh1a/b*) correlated paralogous genes are shown in *Figure 6B–F* and *Figure 6—figure supplement 1*.

However, paralogous pairs that have highly similar temporal patterns and levels may still have spatially distinct expression patterns. To get an estimate of how many of these gene pairs have divergent spatial expression patterns, we used the ZFA annotation provided by ZFIN. There are 683 gene pairs where there is anatomical expression information for both genes. For each pair, we counted up the intersection of all the pairs of stage ID and anatomy term that make up the expression annotations. For the pairs studied, 83.7% (572 pairs) have an intersection of less than 50%. It is important to note that ZFA annotation is not complete, which means that this is an underestimate of the amount of concordance of expression patterns.

GO analysis shows an enrichment for terms related to neural and muscle development (*Supplementary file 11*). Neural terms are already enriched in the whole set of 3144 one-to-one paralogue pairs, but that enrichment becomes even greater when the gene set is limited to the highly correlated paralogue pairs (Pearson coefficient >0.9). By contrast, negatively correlated pairs (Pearson coefficient <−0.5) show limited enrichments. This suggests that temporal divergence is mirrored by greater diversity of GO annotations which in turn leads to reduced term enrichment.

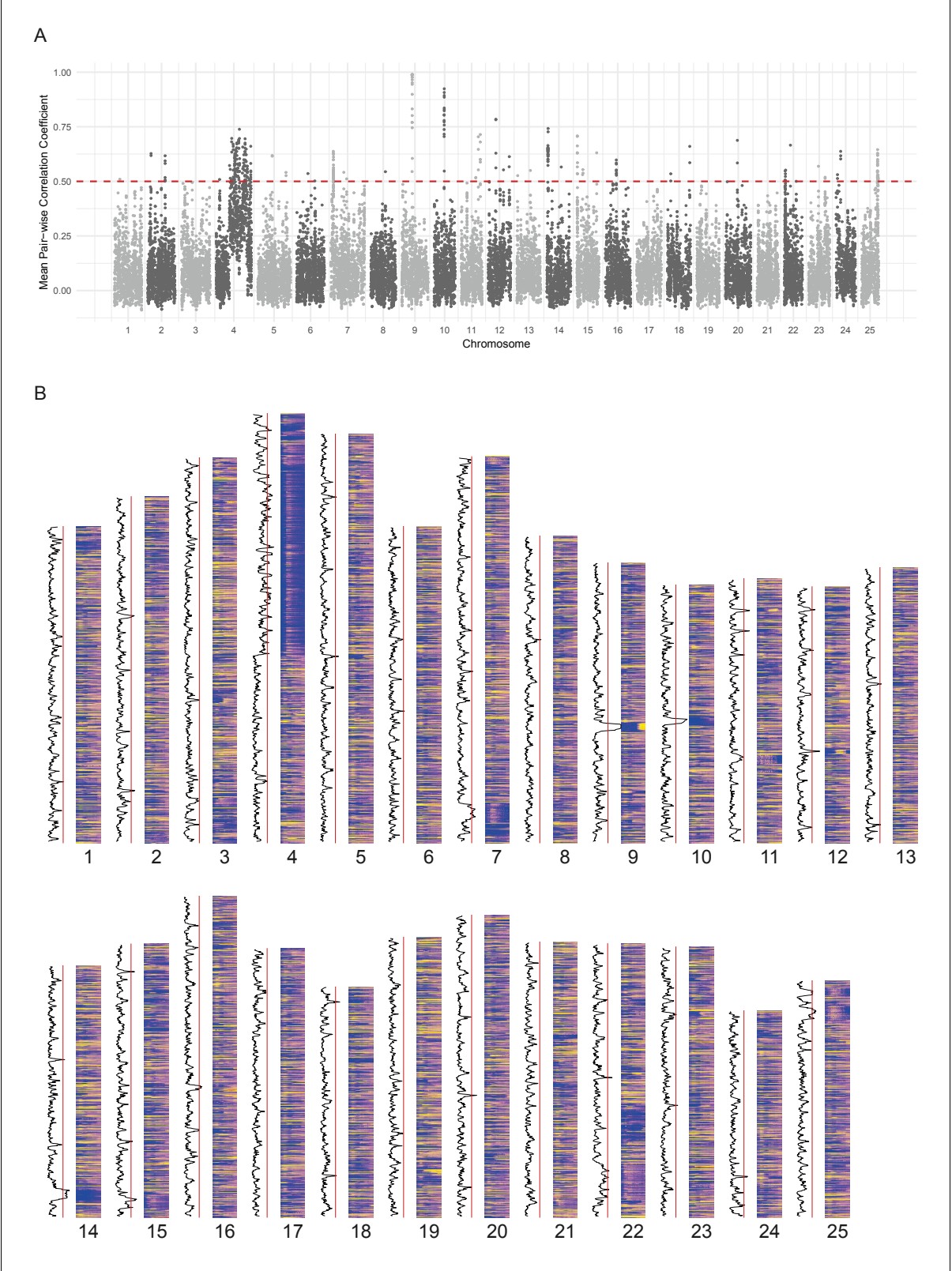

**Figure 5.** Chromosomal expression domains. (**A**) Manhattan plot of the calculated similarity measure (mean pairwise Pearson correlation coefficient) in windows of 10 genes across the genome. (**B**) Heatmaps of the expression profiles (log$_{10}$[TPM]) of genes in chromosomal order. To the left of each histogram, the similarity measure is plotted as a line graph with a red line marking the 0.5 cut-off for identifying regions of co-expression.

DOI: https://doi.org/10.7554/eLife.30860.011

*Figure 5 continued on next page*

*Figure 5 continued*

The following figure supplement is available for figure 5:

**Figure supplement 1.** Heatmaps showing the expression profiles of example regional expression domains.

DOI: https://doi.org/10.7554/eLife.30860.012

## Novel features

An important aspect of a large-scale dataset is the extent to which novel transcripts/genes can be defined. The Ensembl genebuild pipeline contains a procedure to build gene models from RNA-seq data and incorporate these models into the genebuild (*Collins et al., 2012*). Our data have been used by Ensembl to produce an updated gene annotation set, which is available in Ensembl release 90. *Table 2* shows summary statistics comparing release 90 to 89. The new release contains a total of 62,895 transcripts in 35,117 genes. 127 new protein-coding genes (*Supplementary file 12*) and 1166 new transcripts in previously annotated genes have been created as a consequence of incorporating our RNA-seq data (*Figure 7A*). These genes are expressed at lower levels (3-fold) on average as measured by their mean expression across the entire time course (*Figure 7—figure supplement 2A*), which may explain why they haven't previously been annotated. Differential splice isoform usage is a crucial mechanism for generating transcript diversity allowing different forms with potentially divergent functions, localisations and post-transcriptional regulation. To investigate this, we have used Salmon (*Patro et al., 2017*) and maSigPro (*Conesa et al., 2006*; *Nueda et al., 2014*) to produce transcript-level counts for the new e90 genebuild and test for differential isoform usage. We find that 5043 of the 35,117 genes show differential isoform usage across the time course (*Figure 7B*, *Figure 7—figure supplement 1* and *Supplementary files 13–14*) with 3084 genes expressing more than two isoforms across development (*Figure 7C*).

In addition, our data has been used by Ensembl, in conjunction with previously incorporated RNA-seq data, to build 2294 new lincRNAs (*Supplementary file 12*). Of these lincRNAs, 1304 are detected in our dataset with the same cut-off as for protein-coding genes (>0 TPM in all samples for at least one stage) and a high proportion show dynamic expression patterns across the 18 stages (*Figure 7D*). Also, of the detectable lincRNAs, 81 have differential isoform usage during the time course. We compared the Ensembl lincRNAs with two previously published sets (*Pauli et al., 2012*; *Ulitsky et al., 2011*) to investigate the overlap. None of the sets overlap to any great extent, which possibly reflects the different methods and data that were used to identify them (*Figure 7E*). Only 10 of the e90 lincRNAs overlap lincRNAs in both of the other two sets, one of which is *malat1* (ENSDARG00000099970; *Tripathi et al., 2010*; *Ulitsky et al., 2011*). However, all 10 of these were already present in e89.

Like mammalian and previously identified zebrafish lincRNAs (*Cabili et al., 2011*; *Pauli et al., 2012*; *Ulitsky et al., 2011*), they tend to be shorter than protein-coding genes with fewer exons (mean length = 1006 nt, mean number of exons = 3.04; *Figure 7F–G*). They are also expressed at lower levels on average (mean TPM across all samples; protein-coding, median = 7.3 TPM, lincRNA, median = 1.6 TPM; *Figure 7—figure supplement 2B*). Analysis of promoter-associated motifs shows that the transcriptional start sites of these genes are not well-defined (*Figure 7—figure supplement 2C–D*). In mouse and human annotation, the Ensembl lincRNA pipeline and subsequent manual annotation incorporate comprehensive cap analysis of gene expression (CAGE), H3K4me3 and H3K36me3 data for promoter and gene body definition, respectively. By contrast, the zebrafish lincRNA annotation relies on RNA-seq data only, therefore the inherent 3' bias of the expression data leads to less reliable 5' end annotation.

## Novel primary miRNA transcripts

Another class of important transcripts are primary microRNAs (pri-miRNAs). As in other systems, miRNAs play important roles during zebrafish development such as the miR-430 family which is required for proper morphogenesis at early stages and also oriented cell divisions in the neural rod (*Giraldez et al., 2005*; *Takacs and Giraldez, 2016*).

Mature miRNAs (miRNAs) are released from the longer pri-miRNAs by a series of processing steps (*Denli et al., 2004*; *Gregory et al., 2004*; *Grishok et al., 2001*; *Hutvágner et al., 2001*). First, precursor hairpins (pre-miRNAs) are sliced from the host transcript in the nucleus by the

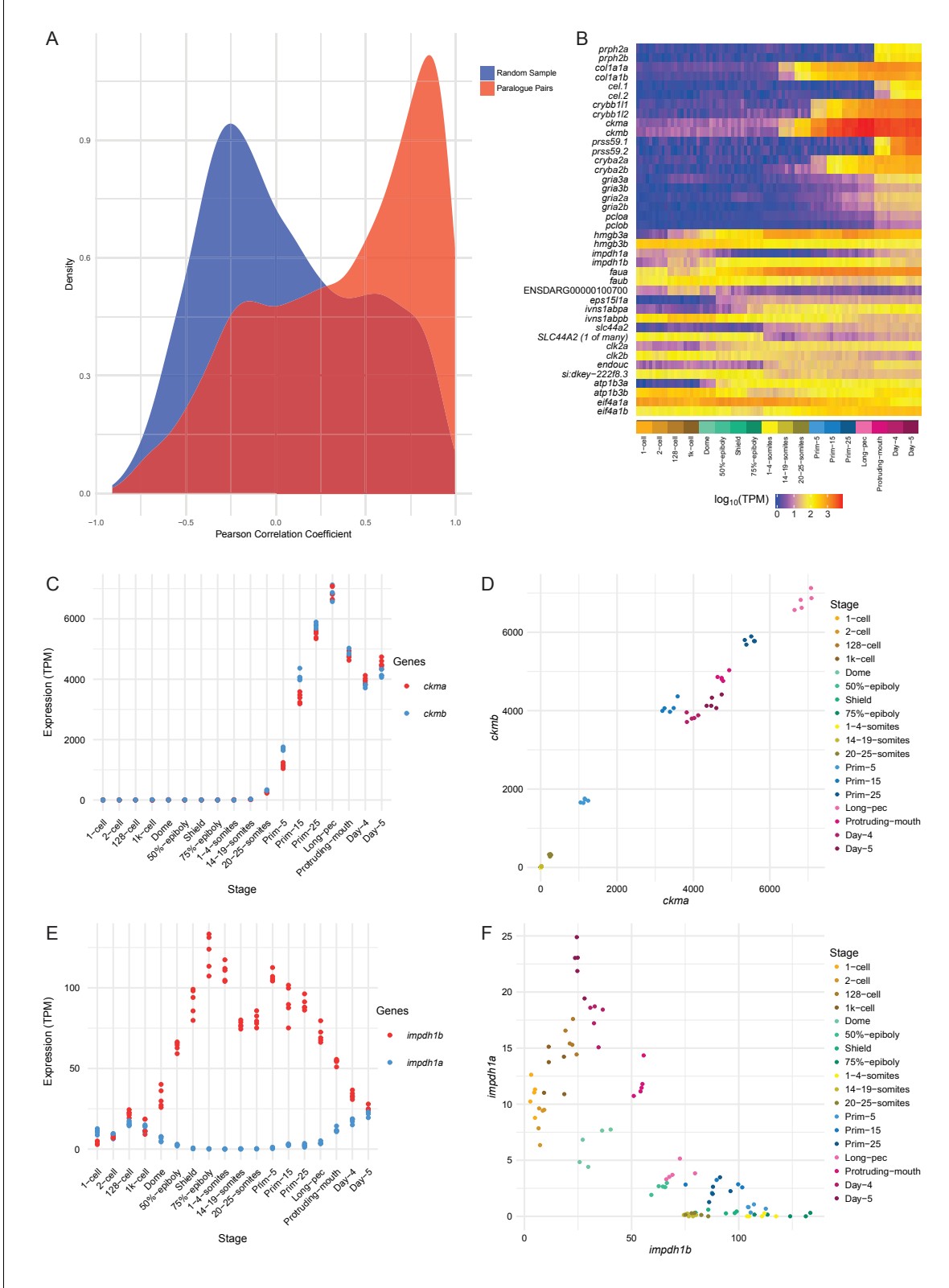

**Figure 6.** Conservation/divergence of paralogous genes. (**A**) Plot of the distribution of Pearson correlation coefficients among one-to-one paralogous gene pairs (red) and a random sample of gene pairs of the same size (blue). (**B**) Heatmap of the expression of the gene pairs with the top 10 highest correlation coefficients (top half) and the top 10 most negative correlation coefficients (bottom half). (**C–D**) Plots displaying the expression of *ckma/b* by stage (**C**) and plotted against each other (**D**). (**E–F**) Plots showing the expression of *impdh1a/b* by stage (**E**) and plotted against each other (**F**).
*Figure 6 continued on next page*

*Figure 6 continued*

DOI: https://doi.org/10.7554/eLife.30860.013

The following figure supplement is available for figure 6:

**Figure supplement 1.** Divergent expression of paralogous pairs.

DOI: https://doi.org/10.7554/eLife.30860.014

microprocessor complex, consisting of the proteins Drosha (Rnasen) and Dgcr8 (Pasha). Subsequently the hairpin is cut by Dicer1 in the cytoplasm, freeing the miRNA duplex.

Although much effort has been dedicated to sequencing and annotating the mature miRNAs themselves, the annotation of many primary miRNA genes is likely hindered by their rapid nuclear processing (*Chang et al., 2015*). Reasoning that the depletion of *drosha* and *dgcr8* may enrich the transcriptome for unprocessed pri-miRNAs, we sequenced RNA derived from *dgcr8* and *drosha* wild-type, heterozygous and homozygous mutant embryos (5 dpf) to determine whether this would allow us to improve the current pri-miRNA annotation.

Following the assembly of the transcriptome, we were able to better define a large number of novel pri-miRNA loci not annotated in Ensembl release v87. The v87 release includes 313 pri-miRNA transcripts overlapping 220 annotated miRNA loci. Our assembly contains 398 pri-miRNA transcripts which overlap 283 miRNA loci, 94 of which are not currently associated with a pri-miRNA (*Figure 8A*, *Supplementary file 15–16*). As with protein-coding genes, there can be multiple different spliced isoforms often having different transcription start sites (*Figure 8B–C*). These pri-miRNA transcripts are now available to view as a selectable track in Ensembl v90 (*Figure 8D*).

## Alternative 3′ end use

Differential use of 3′ ends is another process that achieves transcript diversity and enables differential regulation. There are multiple possible consequences of alternative 3′ ends, such as different/ extra 3′ UTR sequence (*Figure 9A*). This can provide different binding sites for RNA-binding proteins or regulation by microRNAs. For example, one transcript may contain a microRNA binding site lacking in another, changing post-transcriptional regulation. Other possibilities include different/extra coding sequence at the 3′ end of transcripts, leading to proteins that contain different domains and can therefore fulfil different molecular functions. We have catalogued alternative 3′ end usage across development for the same time points as the RNA-seq using a 3′ pulldown method called DeTCT (*Collins et al., 2015*).

As with the RNA-seq data, samples from the same stage correlate well with each other (*Figure 9—figure supplement 1*). PCA plots show very similar patterns to PCA on the RNA-Seq with tight clustering of samples (*Figure 9—figure supplement 2*). Of the genes that contribute more

**Table 2.** Comparison of Ensembl genebuild versions 89 and 90.

| Biotype | e89 | e90 |
| --- | --- | --- |
| protein_coding | 25,672 | 25,743 |
| snRNA | 1287 | 1287 |
| processed_transcript | 1106 | 1106 |
| lincRNA | 914 | 3208 |
| rRNA | 780 | 1579 |
| miRNA | 767 | 440 |
| antisense_RNA | 696 | 696 |
| snoRNA | 297 | 305 |
| unprocessed_pseudogene | 228 | 228 |
| Other | 519 | 525 |
| Total Genes | 32,266 | 35,117 |
| Total Transcripts | 58,549 | 62,895 |

DOI: https://doi.org/10.7554/eLife.30860.015

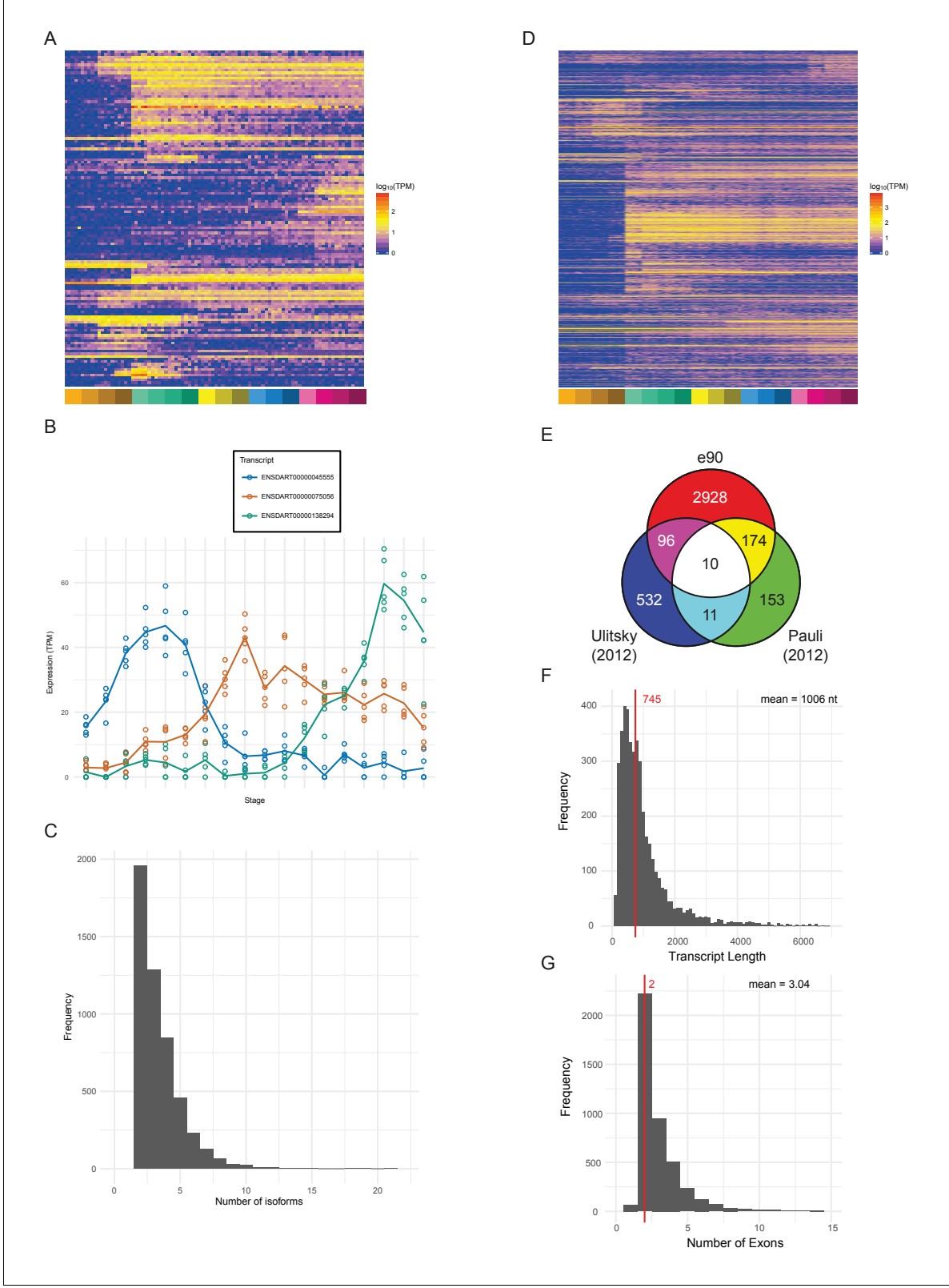

**Figure 7.** Novel features of the dataset. (**A**) Heatmap of the expression profiles (log$_{10}$[TPM]) of the 127 new protein-coding gene annotations in the Ensembl v90 gene build produced by the contribution of this RNA-seq dataset. (**B**) Example of a gene (ENSDARG00000029885, *rab41*) with differential isoform usage across the time course plotted as TPM (points are individual samples and the line represents the mean). (**C**) Histogram of the number of isoforms per gene for genes with differentially expressed isoforms. (**D**) Heatmap of the expression profiles (log$_{10}$[TPM]) of the 1304 detected new

*Figure 7 continued on next page*

*Figure 7 continued*

lincRNA annotations. (E) Venn diagram showing the number of lincRNA regions that overlap with previously published lincRNA sets (*Pauli et al., 2012*; *Ulitsky et al., 2011*). (F) Histogram of the lincRNA transcript lengths. (G) Histogram of the numbers of exons per lincRNA. The red line represents the median and the mean is shown at the top right.

DOI: https://doi.org/10.7554/eLife.30860.016

The following figure supplements are available for figure 7:

**Figure supplement 1.** Differential isoform usage.
DOI: https://doi.org/10.7554/eLife.30860.017
**Figure supplement 2.** Properties of lincRNAs.
DOI: https://doi.org/10.7554/eLife.30860.018

than 0.01% to PC1, 74.6% overlap with those contributing more than 0.01% to PC1 in the RNA-seq PCA (for PC2–6 the percentages are 64.8%, 76.3%, 67.1%, 36.4%, 14.5%; *Supplementary file 17*). This suggests that the largest signals in the data are very similar between the RNA-seq and DeTCT data. Across all 423 samples over all the stages (for each stage there are 11–12 replicates of pools of 8 embryos and 11–12 replicates of single embryos), there were 253,627 regions identified, where each region may be associated with multiple 3′ ends. However, a proportion of these are false positives due to poly(T) pulldown from genomic poly(A) tracts within transcripts. Therefore, we have filtered these regions by various criteria such as distance to an annotated gene and genomic context to remove artefactual regions (*Figure 9B*, *Figure 9—figure supplement 3* and Materials and methods), which reduces the set to 37,724 regions. For this analysis, we increased the stringency of filtering to reduce the inclusion of false-positive ends and produce a high-confidence set of 3′ ends to work from. This stricter set contains 8358 regions. Once the regions have been associated with annotated genes, 1552 genes have two or more 3′ ends (*Supplementary file 18*).

This high-confidence set has allowed us to define 3′ ends of genes that are alternatively used throughout development. For example, *pdlim5b* (ENSDARG00000027600) has three alternative 3′ ends which are used at different times during development (*Figure 9C–D*). During cleavage and gastrula stages, one particular end is used (end 2 in *Figure 9C*), but all three are observed during segmentation stages. Ends 1 and 2 would be predicted to produce the same protein with different 3′ UTRs, allowing for differential post-transcriptional regulation. However, end 3 is predicted to produce a protein which lacks the C-terminal zinc finger domain, allowing for this alternative transcript to have a completely different molecular function (*Figure 9D*).

GO enrichment analysis of the 1552 genes with multiple 3′ ends produces enrichments of a wide variety of biological processes from JNK signalling to exocrine pancreas development with no obvious process or tissue specificity (*Supplementary file 19*). There are no enrichments for Pfam domains, suggesting that there isn't a particular class of gene that is more commonly regulated by alternative 3′ ends.

## Discussion

We have produced a high-quality map of zebrafish mRNA expression during development. It represents the first detailed temporal mRNA reference in zebrafish and enhances our understanding of the transcriptional diversity underlying vertebrate development.

The large number of stages and replicates allows us to cluster genes by their expression profile across all of development using a graph-based method. This splits the assayed genes into 254 different expression profiles containing at least five genes each (11,439 of the genes are assigned to clusters). GO/ZFA enrichment on these clusters allows us to suggest an association of unnamed genes with known processes. There are also many clusters which show no GO enrichments at all, despite having some well-known developmental regulators in them. One possible explanation for this is that genes that already have GO annotations are more likely than unannotated genes to accrue new annotations leading to an inequality in annotation (*Haynes et al., 2017*). This analysis is based on Pearson correlation coefficient between genes across all the samples and so is limited to genes that have a high enough coefficient to be retained and assigned to a cluster. Therefore, it cannot say anything about genes that are not expressed in a similar enough pattern to at least four other genes. Also, since this analysis used whole embryos, it represents a temporal, but not tissue-resolved

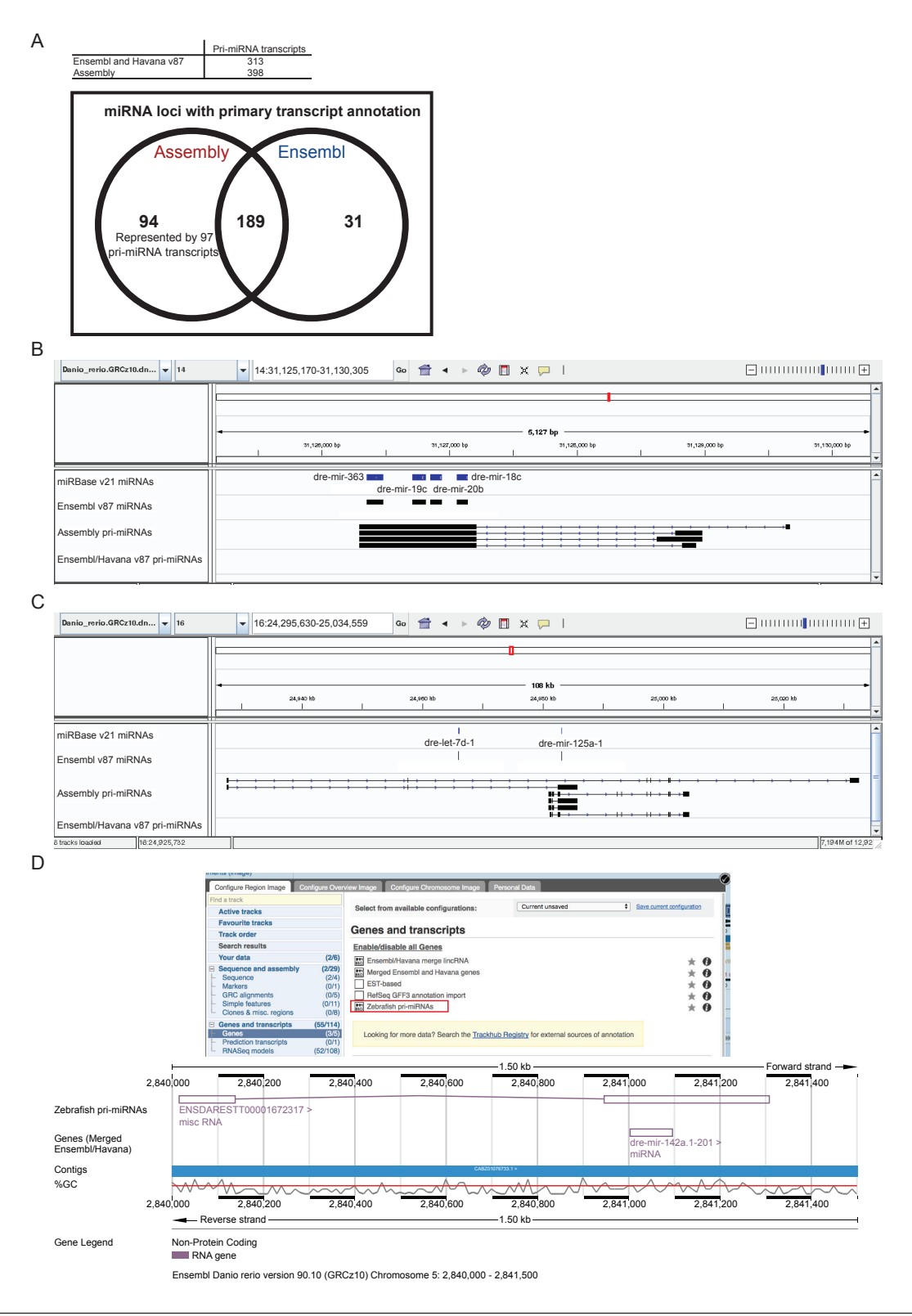

**Figure 8.** Annotation of primary miRNA transcripts. (**A**) Table and Venn diagram of the numbers of assembled pri-miRNA transcripts and the overlap of Ensembl miRNAs with an annotated pri-miRNA. (**B–D**) A selection of miRNA loci for which pri-miRNA annotation is not available in Ensembl and Havana v87. (**B–C**) IGV screen shots illustrating the assembled pri-miRNA transcripts overlapping mir-363, mir-19c, mir-20b and mir-18c (**B**), and mir-125 (**C**). Tracks shown are miRNA annotations from miRBase and Ensembl, newly assembled pri-miRNA transcripts and Ensembl pri-miRNAs. (**D**) Screen

*Figure 8 continued on next page*

*Figure 8 continued*

shots from the Ensembl v90 browser demonstrating how to display the pri-miRNAs. The track is selectable from the 'Genes and transcripts' section of the 'Configure this page' menu. The region around miR-142a (chr5: 2,840,000–2,841,500) is shown with its annotated pri-miRNA.

DOI: https://doi.org/10.7554/eLife.30860.019

overview of development. Studies using tissue- and cell-specific approaches (*Ståhl et al., 2016*; *Trinh et al., 2017*) will refine this baseline with greater resolution and uncover developmental processes in smaller cell populations.

One intriguing set of clusters contains an over-representation of genes on the long arm of chromosome 4, a region considered to be constitutive heterochromatin (*Howe et al., 2013b*). The vast majority of these genes encode proteins with zinc finger domains. The clusters have expression profiles that rise sharply around the time of ZGA and decrease at the latest after 75% epiboly. Given this, it is tempting to speculate that these genes have a role during the activation of the zygotic genome. In *Drosophila* the zinc finger gene Zelda (zld) is involved in zygotic genome activation by early and widespread binding to promoters and enhancers ahead of their genes' activation during the maternal-zygotic transition (*Harrison et al., 2010*; *Harrison et al., 2011*; *Liang et al., 2008*). In zebrafish, *pou5f3*, *nanog* and *soxB* genes have been implicated in activating specific genes during ZGA (*Lee et al., 2013*; *Leichsenring et al., 2013*), but no general factor comparable to Zelda has been identified yet. It is possible that the role of the ZnF protein family genes on chromosome 4 is to set up or maintain open chromatin in a genome-wide way to allow for the rapid activation of the first zygotic genes. This is supported by the finding that these clusters also contain four unnamed genes, also found on chromosome 4, that are predicted to be H3K36 methyltransferases (ENSDARG00000104681, ENSDARG00000091062, ENSDARG00000076160, ENSDARG00000103283). H3K36 methylation is generally, albeit not exclusively, associated with active euchromatin (*Wagner and Carpenter, 2012*). Notably, loss of the chromatin modifier Kdm2aa leads to a specific de-repression of these ZnF genes (*Scahill et al., 2017*).

Another option is that they could be responsible for negatively regulating areas of open chromatin to stop inappropriate activation of transposable elements during ZGA. In mammals, suppression of retrotransposons is mediated by the rapidly evolving KRAB zinc finger genes (*Jacobs et al., 2014*). KRAB zinc finger genes are not found in ray-finned fish (*Imbeault et al., 2017*), making the ZnF genes on chromosome 4 a possible functional equivalent.

Also, as would be expected given their expression profiles, within these four clusters are a large number of well-known developmental regulators such as members of the Tgf-beta (Bmp and Nodal) and Wnt signalling pathways, as well as transcription factors such as *gata5* & *6*, *sox32*, *vox*, *ved*, *vent* and both *Brachyury* homologues.

## Paralogue analysis

Correlation analysis of paralogues stemming from a teleost-specific genome duplication shows that although there is the expected skew towards positive temporal expression correlation, a significant proportion of paralogues have diverged in their expression pattern as demonstrated by low or negative correlation coefficients. This is confirmed by an analysis of the expression annotations provided by ZFIN. However, it is important to note that the spatial-temporal expression annotation is far from complete and, therefore, annotations may not overlap simply because a certain stage has been investigated for one paralogue, but not the other. It will also be the case that some genes have expression patterns that are diverged spatially, but not temporally and we will not necessarily be able to detect these events. Thus, the number of genes whose expression annotations do not overlap is an estimate, but it does suggest that the phenomenon of divergence is widespread among paralogues, something that might be expected to be the case since there needs to be some form of selective pressure on retaining paralogous pairs. GO enrichment of highly correlated paralogues reveals a skew towards neural and muscle expression whereas negatively correlated pairs do not exhibit this expression restriction, which is likely to reflect greater GO term diversity among the latter.

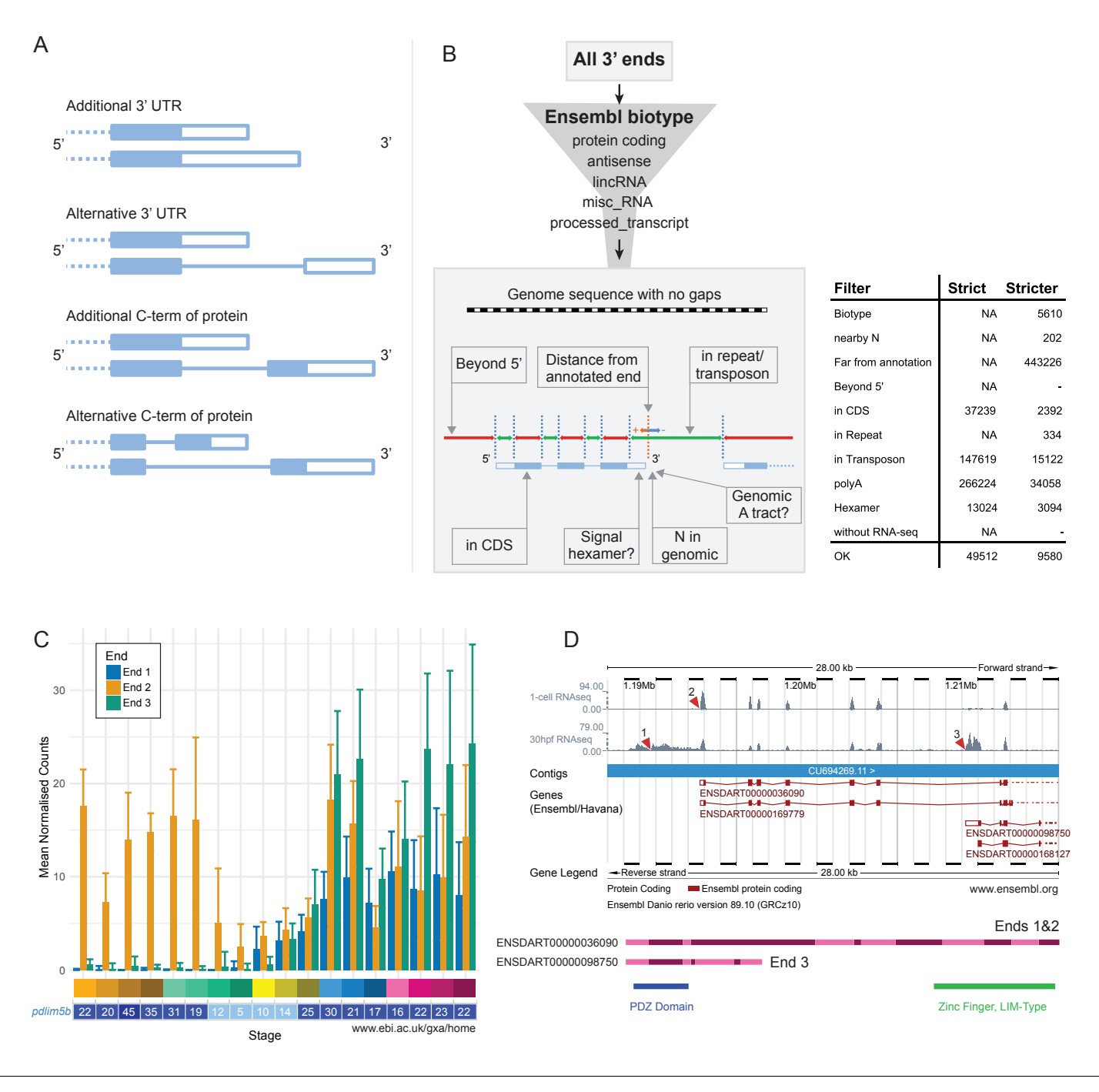

**Figure 9.** Alternative 3′ end use during development. (**A**) Diagrams illustrating the possible outcomes of differential 3′ end usage. Only the 3′ end of the gene is shown, where the exons are boxes, the introns are represented as lines and filled boxes are coding. The dotted line represents the remaining 5′ end of the gene. (**B**) Flowchart of the DeTCT 3′ end filtering strategy. See Materials and methods for the definition of each filter. The table shows the number of ends removed by each filter when using the strict/stricter settings. (**C**) Graph of the mean normalised counts by stage for the three alternative 3′ ends (shown in D) of the *pdlim5b* gene. Error bars represent standard deviations. Underneath is a screenshot from Expression Atlas (Aug 2017) showing the overall TPM for *pdlim5b*. (**D**) Screenshot from Ensembl (release 89) showing the 3′ portion of the *pdlim5b* gene (10:1,187,000–1,215,000). The top two tracks are RNA-seq coverage for 1 cell and prim-15 (30 hpf). Red arrowheads point to the genomic positions of the three transcript ends. Note, the 3′ UTR resulting in end 1 is not annotated in this genebuild. Below is a schematic of the proteins that are predicted to be produced by two of the transcripts. The positions of domains within the protein are also indicated.

DOI: https://doi.org/10.7554/eLife.30860.020

The following figure supplements are available for figure 9:

*Figure 9 continued on next page*

*Figure 9 continued*

**Figure supplement 1.** DeTCT QC.
DOI: https://doi.org/10.7554/eLife.30860.021
**Figure supplement 2.** DeTCT PCA.
DOI: https://doi.org/10.7554/eLife.30860.022
**Figure supplement 3.** 3' end filtering.
DOI: https://doi.org/10.7554/eLife.30860.023

## Novel genes

Our data has added 127 previously unknown potential protein-coding genes to Ensembl's annotation. These genes are expressed at lower levels on average than the entire set of protein-coding genes and many are expressed in a limited number of stages. This may explain why there has not been enough evidence previously for Ensembl's gene-build pipeline to annotate them. Similarly, 2294 new genes annotated as lincRNAs have been added. This compares with 7821 currently annotated in Ensembl for the human genome. Properties of the lincRNAs in terms of numbers of exons, length of transcript and overall expression level are similar to what has been reported previously (*Cabili et al., 2011*; *Pauli et al., 2012*; *Ulitsky et al., 2011*). The fact that there is so little overlap between the three sets is probably because of the different methods that were used to define them. *Ulitsky et al. (2012)* used H3K4me3 peaks and 3'-end profiling to define regions of transcription, and transcript models were built within these regions. *Pauli et al. (2012)* used RNA-seq data to assemble a set of transcripts which were then filtered for coding potential using PhyloCSF. The Ensembl lincRNA pipeline uses the RNA-seq models that do not overlap protein-coding genes and filters them by looking for Pfam (*Finn et al., 2014*) domains in all candidate protein sequences in all frames in the transcript. The numbers of transcripts produced by the Ensembl pipeline suggest that it is more sensitive, but less specific than the other two methods. For human and mouse genomes, Ensembl uses chromatin methylation state (H3K4me3 and H3K36me3), but does not yet use this for the zebrafish. Incorporation of CAGE data would also help to correctly identify the 5' ends of these transcripts.

The involvement of microRNAs during embryonic development is widespread (reviewed in *Bartel, 2009*; *Mishima, 2012*). However, the primary transcripts of miRNAs are often poorly annotated. Using two miRNA processing mutants (*dgcr8* and *drosha*), we have produced an improved primiRNA assembly with an increase of 42% in the number of Ensembl miRNAs with an associated primary transcript, including genes implicated in developmental processes in zebrafish (*Figure 9D*, *Fan et al., 2014*). In agreement with previous studies (*Chang et al., 2015*; *Gaeta et al., 2017*), the assembly shows that miRNAs are processed from primary transcripts with multiple isoforms with potentially multiple transcription start sites (*Figure 9*). This diversity of transcripts means that the expression of miRNAs in relatively close proximity (e.g. dre-let-7d-1 and dre-mir-125a-1) could be regulated at multiple levels (*Figure 9C*). In some transcripts let-7d-1 and mir-125a-1 are both intronic and in some mir-125a-1 is exonic. There are also transcripts with alternate promoters which exclude the let-7d-1 locus entirely. These features provide significant flexibility to the transcriptional regulation of miRNAs. It is possible that a proportion of the assembled constructs are fragments. Clear definition of the 5' and 3' termini will require further work and ultimately experimental validation by an alternative method such as 5' rapid amplification of cDNA ends.

## Transcript diversity

A further layer of complexity is added by differential exon and 3' end use. The dataset uncovers new splice isoforms that show differential expression across the 18 stages. When viewed in Ensembl against the Ensembl genebuild, this refines the current gene annotation and ties splice isoforms to their temporal expression patterns. This is of particular relevance to the ongoing debate surrounding reverse genetics approaches in zebrafish (*Kok et al., 2015*). One possible explanation for phenotypic discrepancies between morpholino antisense oligo knockdown, code disrupting point mutations and CRISPR/Cas9 or TALEN-mediated larger deletions is that mutations affect potentially noncritical exons that are only present in a subset of transcripts of a given gene. Indeed, recent publications demonstrate that genes carrying code-disrupting mutations created by ENU or CRISPR/Cas9

mutagenesis can produce potentially functional transcripts by exon skipping (*Anderson et al., 2017*; *Mou et al., 2017*). An improved annotation that gives temporal resolution of exon usage is of great value for the identification of critical exons.

Using the 3′ end sequencing method DeTCT, we have catalogued the differential use of alternative 3′ ends in a filtered set of transcripts. In accordance with previous work on a subset of stages (*Collins et al., 2012*; *Sheppard et al., 2013*; *Ulitsky et al., 2012*) we find that there is differential use of not only 3′ UTRs but also coding sequence across development. We show that *pdlim5b* is expressed as several transcripts, with a single maternal isoform and multiple zygotic ones which would be predicted to produce different proteins. This type of data provides scope for temporally resolved analysis using transcript-specific knockout targeting strategies.

A major aim for this work was to create an accessible reference of normal mRNA expression during zebrafish development. The primary sequence is available from the European Nucleotide Archive (ENA, http://www.ebi.ac.uk/ena), but importantly has also been interpreted for immediate user-friendly access. Expression Atlas (http://www.ebi.ac.uk/gxa/experiments/E-ERAD-475/Results) provides relative expression levels and Ensembl displays alternative genomic transcript structures for each stage alongside the current gene annotation. These pre-computed analyses allow an in-depth examination of the data on a gene by gene basis. This makes it easy for researchers to benefit from the data and provides a direct link to the wealth of information present in genomic databases.

# Materials and methods

## Key resource table

| Reagent type (species) or resource | Designation | Source or reference | Identifiers | Additional information |
|---|---|---|---|---|
| genetic reagent (*Danio rerio*) | *drosha*^sa191 | this paper | ZFIN:sa191; RRID: ZFIN_ZDB-ALT-100831-1 | ENU mutant. C - > T at position GRCz10:12:13311049. Available from ZIRC (http://zebrafish.org/fish/lineAll.php?OID=ZDB-ALT-100831-1) |
| genetic reagent (D. rerio) | *dgcr8*^sa223 | this paper | ZFIN:sa223; ZFIN: ZDB-ALT-161003–11399 | ENU mutant. C - > T at position GRCz10:12:24021119 |
| strain, strain background (D. rerio) | HLF | this paper | | Hubrecht Long Fin |
| software, algorithm | TopHat | 10.1038/nprot.2012.016 | RRID:SCR_013035 | |
| software, algorithm | Cufflinks | 10.1038/nprot.2012.016 | RRID:SCR_014597 | |
| software, algorithm | QoRTs | 10.1186/s12859-015-0670-5 | | https://github.com/hartleys/QoRTs |
| software, algorithm | htseq-count | 10.1093/bioinformatics/btu638 | RRID:SCR_011867 | |
| software, algorithm | R | https://www.r-project.org/ | RRID:SCR_001905 | |
| software, algorithm | Perl | https://perldoc.perl.org/ | | |
| software, algorithm | zfs-devstages | this paper | | github repository, https://github.com/richysix/zfs-devstages |
| software, algorithm | Ontologizer | 10.1093/bioinformatics/btn250 | RRID:SCR_005801 | |
| software, algorithm | STAR | 10.1093/bioinformatics/bts635 | | |
| software, algorithm | Samtools | 10.1093/bioinformatics/btp352 | RRID:SCR_002105 | |
| software, algorithm | Picard | http://broadinstitute.github.io/picard | RRID:SCR_006525 | |
| software, algorithm | gtfToGenePred | UCSC Genome Browser | | http://hgdownload.soe.ucsc.edu/downloads.html#utilities_downloads |
| software, algorithm | genePredToBed | UCSC Genome Browser | | http://hgdownload.soe.ucsc.edu/downloads.html#utilities_downloads |

## Animal care

Zebrafish were maintained in accordance with UK Home Office regulations, UK Animals (Scientific Procedures) Act 1986, under project licence 70/7606, which was reviewed by the Wellcome Trust Sanger Institute Ethical Review Committee.

## Sample collection

Wild-type HLF strain zebrafish (*Danio rerio*) were maintained at 23.5°C on a 14 hr light/10 hr dark cycle. At the time of mating, breeding males and females were separated overnight before letting them spawn naturally in the morning to allow for synchronisation of developmental stages. Fertilised eggs were grown at 28°C and staged using previously defined criteria (*Kimmel et al., 1995*). Samples from 18 different developmental stages from 1 cell to 5 days post-fertilisation were collected by snap freezing the embryos in dry ice. Details of sample names and accession numbers are provided in *Supplementary file 1*.

## RNA extraction

RNA was extracted from embryos by mechanical lysis in RLT buffer (Qiagen) containing 1 µl of 14.3M beta mercaptoethanol (Sigma). The lysate was combined with 1.8 volumes of Agencourt RNAClean XP (Beckman Coulter) beads and allowed to bind for 10 min. The plate was applied to a plate magnet (Invitrogen) until the solution cleared and the supernatant was removed without disturbing the beads. This was followed by washing the beads three times with 70% ethanol. After the last wash, the pellet was allowed to air dry for 10 mins and then resuspended in 50 µl of RNAse-free water. RNA was eluted from the beads by applying the plate to the magnetic rack. RNA was quantified using Quant-IT RNA assay (Invitrogen) for samples 24 hr post-fertilisation and older.

## RNA sequencing

Total RNA from 12 embryos was pooled and DNase treated for 20 mins at 37°C followed by addition of 1 µl 0.5M EDTA and inactivation at 75°C for 10 mins to remove residual DNA. RNA was then cleaned using 2 volumes of Agencourt RNAClean XP (Beckman Coulter) beads under the standard protocol. Five replicate libraries were made for each stage using 700 ng total RNA and ERCC spike mix 2 (Ambion). Strand-specific RNA-seq libraries containing unique index sequences in the adapter were generated simultaneously following the dUTP method. Libraries were pooled and sequenced on Illumina HiSeq 2500 in 100 bp paired-end mode. Sequence data were deposited in ENA under accession ERP014517.

FASTQ files were aligned to the GRCz10 reference genome using TopHat (v2.0.13, options: –library-type fr-firststrand, *Kim et al., 2013*). The data were assessed for technical quality (GC-content, insert size, proper-pairs etc.) using QoRTs (*Hartley and Mullikin, 2015*). Counts for genes were produced using htseq-count (v0.6.0 options: –stranded=reverse, *Anders et al., 2015*) with the Ensembl v85 annotation as a reference. The processed count data are in *Supplementary file 2*.

## Analysis

Count data was converted to Transcripts per Million (TPM) and all subsequent analysis was done using TPM (*Supplementary file 3*). Analysis was done using R (*R Core Team, 2015*) and Perl as well as the tools mentioned below. The code to reproduce all the plots in this paper can be downloaded from GitHub (https://github.com/richysix/zfs-devstages) (*White, 2017*). A copy is archived at https://github.com/elifesciences-publications/zfs-devstages.

## Sample correlation matrix

The sample correlation matrix was produced by calculating the Spearman correlation coefficient between each possible pair of samples. Spearman correlation coefficient was used in preference to Pearson because it is less sensitive to outliers.

## Principal component analysis

Principal Component Analysis was carried out using the prcomp function in R. This projects the data down from 32,105 dimensions (genes) to 90 (samples). Each component accounts for a fraction of the variance present in the data with the first six components together capturing over 90%. The genes that contribute most to each component were determined by calculating the proportion of variance explained by each gene for each component and selecting those above 0.01%.

## ZFA enrichment

The ZFA (*Van Slyke et al., 2014*) is a controlled vocabulary for describing the anatomy and development of zebrafish and is used by the Zebrafish Information Network (ZFIN, *Howe et al., 2013a*) to annotate the spatial expression patterns of genes from the literature and large-scale whole mount mRNA in situ hybridisation experiments (*Rauch et al., 2003*; *Thisse et al., 2001*). ZFA enrichment was done with Ontologizer (http://ontologizer.de, *Bauer et al., 2008*) using the 'Parent-Child-Union' calculation method and Benjamini-Hochberg correction for multiple testing. The ZFA ontology was downloaded from http://www.berkeleybop.org/ontologies/zfa.obo and mappings between ZFIN gene IDs and ZFA IDs were downloaded from ZFIN (http://zfin.org/downloads/wildtype-expression_fish.txt and http://zfin.org/downloads/phenoGeneCleanData_fish.txt).

## GO enrichment

GO enrichment was performed using the R topGO package (*Alexa and Rahnenfuhrer, 2016*). The mapping between Ensembl IDs and GO terms was retrieved from the Ensembl database using a custom Perl script (get_ensembl_go_terms.pl) from the topgo-wrapper repository (https://github.com/iansealy/topgo-wrapper).

## Unnamed genes

Unnamed genes were defined by matching gene names to the following patterns: [A-Z][A-Z][0–9]{6}\.[0–9]+, CABZ[0–9]{8}\.[0–9]+, im:[0–9]{7}, si:.*, wu:.*, zgc:[0–9]{5,6}. Orthologues (along with the predicted last common ancestor) were obtained from the Ensembl Compara database (*Vilella et al., 2009*) using a custom Perl script (https://github.com/richysix/zfs-devstages) for the following species: *Astyanax mexicanus*, *Gasterosteus aculeatus*, *Oryzias latipes*, *Takifugu rubripes*, *Tetraodon nigroviridis*, *Gadus morhua*, *Mus musculus*, *Rattus norvegicus* and *Homo sapiens*.

Genes with no identifiable orthologue in any of the species were classed as *Danio*-specific. Those with orthologues in the six fish species but not the mammals examined (last common ancestor Neopterygii, Clupeocephala or Otophysi) were categorised as Teleost-specific. Those with orthologues also present in the mammalian species (last common ancestor Euteleostomi, Vertebrata, Chordata or Bilateria) were labelled as Vertebrate.

## BioLayout

BioLayout *Express*3D is available to download for free (http://www.biolayout.org/downloads/). The input to the program is the expression values for the genes in each sample. The Pearson correlation cut-off was set to 0.94; gene pairs with a correlation coefficient above this are linked by an edge in the input graph (13162 genes). After running MCL, clusters with fewer than six genes were removed (1723 genes) which produced 254 clusters ranging in size from 6 to 2360 genes. The input expression file is provided as *Supplementary file 4*. The processed layout file of the network displayed in *Figure 3B* is available at https://doi.org/10.6084/m9.figshare.4622293.

## Chromosome enrichment in BioLayout clusters

Enrichment for particular chromosomes within BioLayout clusters was tested for using a one-tailed binomial test followed by multiple testing correction using the R qvalue package (*Storey, 2003*; *Storey et al., 2015*) at FDR = 0.05. For each cluster, only chromosomes that appeared more than five times were tested.

## Chromosomal co-expression regions

To calculate a similarity measure for groups of genes in a chromosomal region the genome was split up into overlapping windows of 10 genes. For each window, the Pearson correlation coefficient was calculated for every combination of pairs of genes (45 pairs for a group of 10 genes) and averaged. Windows with a mean correlation coefficient greater than 0.5 were selected and overlapping windows were merged.

## Paralogous gene pairs

One-to-one paralogues were identified from the Ensembl Compara database (*Vilella et al., 2009*) using a custom Perl script (https://github.com/richysix/zfs-devstages) filtering for one-to-one

relationships and ones where the last common ancestor was one of Clupeocephala, Danio rerio, Neopterygii or Otophysi to get paralogous pairs arising from the most recent whole genome duplication event (*Meyer and Schartl, 1999*).

ZFA annotation of wild-type gene expression data was downloaded from ZFIN (http://zfin.org/downloads/wildtype-expression_fish.txt). For each paralogous pair, the overlap in ZFA term, ZFS stage pairs was calculated. Image data for this paper were retrieved from the Zebrafish Information Network (ZFIN), University of Oregon, Eugene, OR 97403–5274; URL: http://zfin.org/; 23 Jan 2017.

## Differential isoform usage

Transcript level counts were produced using Salmon (*Patro et al., 2017*) in Quasi-mapping-based mode with the new Ensembl e90 transcripts as a reference (v0.8.2, options: –threads 4 –seqBias –gcBias –libType ISR). Differentially spliced genes and differentially expressed transcripts were called using the R package maSigPro (*Conesa et al., 2006*; *Nueda et al., 2014*) after removing the 20% of genes with the lowest expression.

## DeTCT

DeTCT libraries were generated, sequenced and analysed as described previously (*Collins et al., 2015*). The resulting genomic regions and putative 3′ ends were filtered using DeTCT's filter_output script (https://github.com/iansealy/DETCT/blob/master/script/filter_output.pl) in both its –strict and –stricter modes. –stricter mode removes 3′ ends more than 5000 bases downstream of existing protein-coding annotation (Ensembl v86) or more than 50 bases from existing non-coding annotation (antisense, lincRNA, misc_RNA and processed_transcript biotypes). 3′ ends are also removed if nearby sequence is enriched for As, if within 14 bp of an N, if within coding sequence, a simple repeat (repeats annotated by Dust or TRF) or a transposon, or if not near a primary hexamer. –strict mode removes 3′ ends in coding sequence, transposons, if nearby sequence is enriched for As or if not near a primary hexamer. Regions not associated with 3′ ends are also removed. Sequence data were deposited in ENA under accessions ERP006948 and ERP013756. The processed count data is available to download at https://doi.org/10.6084/m9.figshare.4622311.

## pri-miRNA assembly

For each mutant allele – *drosha*[sa191] (*rnasen*); *dgcr8*[sa223] (*pasha*) – embryos were collected at 5 dpf and genotyped by KASP genotyping as previously described (*Dooley et al., 2013*). We produced and sequenced stranded RNA-seq libraries from single homozygous, heterozygous and wild-type embryos (5 replicates of each). Sequence data were deposited in ENA under accession ERP013690. To identify new pri-miRNAs, FASTQ files were aligned to the zebrafish genome (GRCz10) using STAR (*Dobin et al., 2013*) v2.5.1a, options: –alignSJoverhangMin 8 –alignSJDBoverhangMin 1 –outFilterMismatchNmax 999 –alignIntronMin 20 –alignIntronMax 1000000 –alignMatesGapMax 1000000 –readFilesCommand zcat –outFilterIntronMotifs RemoveNoncanonical –outFilterType BySJout –outFilterMultimapNmax 10 –sjdbOverhang 74. Ensembl v86 zebrafish annotation was provided as a junction file. Aligned reads were sorted, indexed and mate pairings were fixed using Samtools (v1.3.1, *Li et al., 2009*). Duplicate alignments were removed with Picard (http://broadinstitute.github.io/picard, v2.7.1, REMOVE_DUPLICATES = true READ_NAME_REGEX = null). Alignments for the homozygous, heterozygous and wild-type samples for both the *drosha* and *dgcr8* lines were merged into two line-specific datasets with Samtools. Each set of merged alignments was then used to assemble a transcriptome with Cufflinks (v2.2.1, *Trapnell et al., 2012*) (–library-type=fr firststrand). Ensembl (v86) annotation stripped of rows corresponding to 'gene' features was provided as a backbone for the assembly. The *drosha* and *dgcr8* assemblies were then merged with Cuffmerge.

Ensembl miRNA annotation (v87) ('miRNA' biotype) was used to define miRNA loci. For the purpose of this analysis we defined pri-miRNAs as any spliced transcript overlapping at least one of these loci. Ensembl and Havana (v87) annotation was selected as the baseline for novel pri-miRNA prediction. UCSC utilities gtfToGenePred and genePredToBed were used to convert from GTF to BED format. Pri-miRNAs were identified by removing single exon transcripts from the merged assembly and Ensembl annotations and comparing the remainder to annotated miRNA loci using bedtools (v2.24.0) intersect (-wa –s). Finally, assembled pri-miRNA transcripts were filtered to

remove those for which either the first or last exon was less than 20nt in length, those which contain an intron greater than 100,000 nt, or those which contain an intron that isn't flanked by the GT-AG or GC-AG motifs (*Breathnach et al., 1978*).

## Ensembl lincRNA pipeline

lincRNAs are annotated by a separate pipeline that is run after the RNA-seq and genebuild pipelines. In the first stage, this method traverses each toplevel sequence (e.g. chromosomes) and collects the models created by the RNA-seq pipeline that neither overlap known protein-coding genes nor were used to build protein-coding genes. These potential non-coding models are filtered by looking for Pfam (*Finn et al., 2014*) domains in all candidate protein sequences (and frames) to determine the coding potential of the model. Finally, if the RNA-seq model is long enough and doesn't have coding potential, it is labelled as a lincRNA.

## Acknowledgements

We would like to thank the Wellcome Trust Sanger Institute sequencing pipelines for performing sequencing and the staff of the Research Support Facility for zebrafish care. We also thank Samantha Carruthers, Catherine Scahill and Nicole Staudt for critical reading of the manuscript and Ian Dunham for helpful discussions.

## Additional information

### Funding

| Funder | Grant reference number | Author |
| --- | --- | --- |
| Wellcome | WT098051,206194 | Richard J White<br>John E Collins<br>Ian M Sealy<br>Neha Wali<br>Christopher M Dooley<br>Zsofia Digby<br>Derek L Stemple<br>Elisabeth M Busch-Nentwich |
| European Molecular Biology Laboratory | | Daniel N Murphy<br>Konstantinos Billis<br>Thibaut Hourlier<br>Anja Füllgrabe<br>Anton J Enright |
| National Institutes of Health | 1R01HD074078 | Daniel N Murphy |
| Wellcome | WT108749/Z/15/Z | Daniel N Murphy<br>Konstantinos Billis<br>Thibaut Hourlier |
| Medical Research Council | MR/L012367/1 | Matthew P Davis |
| Biotechnology and Biological Sciences Research Council | BB/J01589X/1 | Anton J Enright |

The funders had no role in study design, data collection and interpretation, or the decision to submit the work for publication.

### Author contributions

Richard J White, Data Curation, Formal analysis, Investigation, Software, Visualization, Writing—original draft, Writing—review and editing; John E Collins, Conceptualization, Investigation, Visualization, Writing—review and editing; Ian M Sealy, Formal Analysis, Software, Writing—review and editing; Neha Wali, Christopher M Dooley, Zsofia Digby, Investigation; Derek L Stemple, Funding Acquisition; Daniel N Murphy, Formal Analysis; Konstantinos Billis, Formal analysis, Software; Thibaut Hourlier, Anja Füllgrabe, Anton J Enright, Formal analysis; Matthew P Davis, Formal analysis, Visualization, Writing—original draft; Elisabeth M Busch-Nentwich, Conceptualization, Funding acquisition, Supervision, Visualization, Writing—original draft, Writing—review and editing

### Author ORCIDs
Richard J White http://orcid.org/0000-0003-1842-412X
Konstantinos Billis http://orcid.org/0000-0001-8568-4306
Thibaut Hourlier http://orcid.org/0000-0003-4894-7773
Anja Füllgrabe http://orcid.org/0000-0002-8674-0039
Matthew P Davis https://orcid.org/0000-0002-9909-3651
Elisabeth M Busch-Nentwich http://orcid.org/0000-0001-6450-744X

### Ethics

Animal experimentation: Zebrafish were maintained in accordance with UK Home Office regulations, UK Animals (Scientific Procedures) Act 1986, under project licence 70/7606, which was reviewed by the Wellcome Trust Sanger Institute Ethical Review Committee.

### Decision letter and Author response

Decision letter https://doi.org/10.7554/eLife.30860.050
Author response https://doi.org/10.7554/eLife.30860.051

## Additional files

### Supplementary files

• Supplementary file 1. Sample Information (.tsv)
DOI: https://doi.org/10.7554/eLife.30860.024

• Supplementary file 2. RNA-seq count data (.tsv)
DOI: https://doi.org/10.7554/eLife.30860.025

• Supplementary file 3. RNA-seq TPM (.tsv)
DOI: https://doi.org/10.7554/eLife.30860.026

• Supplementary file 4. BioLayout input file (.expression)
DOI: https://doi.org/10.7554/eLife.30860.027

• Supplementary file 5. BioLayout cluster information (.tsv)
DOI: https://doi.org/10.7554/eLife.30860.028

• Supplementary file 6. BioLayout cluster GO/ZFA enrichment (.zip)
DOI: https://doi.org/10.7554/eLife.30860.029

• Supplementary file 7. undetected genes GO enrichment (.tsv)
DOI: https://doi.org/10.7554/eLife.30860.030

• Supplementary file 8. Cluster7, 23, 26, 135 Pfam info (.tsv)
DOI: https://doi.org/10.7554/eLife.30860.031

• Supplementary file 9. Regional expression information (.zip)
DOI: https://doi.org/10.7554/eLife.30860.032

• Supplementary file 10. Paralogue Information (.tsv)
DOI: https://doi.org/10.7554/eLife.30860.033

• Supplementary file 11. Paralogue GO enrichment (.tsv)
DOI: https://doi.org/10.7554/eLife.30860.034

• Supplementary file 12. new genes (protein-coding and lincRNAs) in Ensembl v90 (.bed)
DOI: https://doi.org/10.7554/eLife.30860.035

• Supplementary file 13. e90 transcript level TPM (.tsv)
DOI: https://doi.org/10.7554/eLife.30860.036

• Supplementary file 14. Differential isoform usage plots (.pdf)
DOI: https://doi.org/10.7554/eLife.30860.037

• Supplementary file 15. Assembled pri-miRNA transcripts (.bed)
DOI: https://doi.org/10.7554/eLife.30860.038

• Supplementary file 16. Table of overlap of miRNAs (.xlsx)

DOI: https://doi.org/10.7554/eLife.30860.039

• Supplementary file 17. Genes contributing to Principal Components (RNAseq and DeTCT) (.zip)
DOI: https://doi.org/10.7554/eLife.30860.040

• Supplementary file 18. Genes with multiple 3' ends (.tsv)
DOI: https://doi.org/10.7554/eLife.30860.041

• Supplementary file 19. GO enrichment - Genes with multiple 3' ends (.tsv)
DOI: https://doi.org/10.7554/eLife.30860.042

• Transparent reporting form
DOI: https://doi.org/10.7554/eLife.30860.043

## Major datasets

The following datasets were generated:

| Author(s) | Year | Dataset title | Dataset URL | Database, license, and accessibility information |
| --- | --- | --- | --- | --- |
| Collins JE, Wali N, Dooley CM, Busch-Nentwich EM | 2016 | Baseline_expression_from_transcriptional_profiling_of_zebrafish_developmental_stages_2 | http://www.ebi.ac.uk/ena/data/view/PRJEB12296 | Publicly available at the EBI European Nucleotide Archive (accession no: PRJEB12296) |
| Collins JE, Wali N, Dooley CM, Busch-Nentwich EM | 2016 | Baseline_expression_from_transcriptional_profiling_of_zebrafish_developmental_stages | http://www.ebi.ac.uk/ena/data/view/PRJEB7244 | Publicly available at the EBI European Nucleotide Archive (accession no: PRJEB7244) |
| Collins JE, Wali N, Dooley CM, Busch-Nentwich EM | 2016 | Baseline_expression_from_transcriptional_profiling_of_zebrafish_developmental_stages_3 | http://www.ebi.ac.uk/ena/data/view/PRJEB12982 | Publicly available at the EBI European Nucleotide Archive (accession no: PRJEB12982) |

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
