## [Decision Letter]

Thank you for submitting your article "A high-resolution mRNA expression time course of embryonic development in zebrafish" for consideration by *eLife*. Your article has been reviewed by three peer reviewers, and the evaluation has been overseen by Didier Stainier as the Senior Editor and Reviewing Editor. The following individuals involved in review of your submission have agreed to reveal their identity: Shawn Burgess (Reviewer #1) and Tatjana Sauka-Spengler (Reviewer #3).

The reviewers have discussed the reviews with one another and the editor has drafted this decision to help you prepare a revised submission.

Summary:

This manuscript represents a large effort to generate a major genomic resource for the zebrafish community. The work comprises quality, strand-specific RNA-seq across 18 early developmental time points. The quality and depth of sequencing identified over 100 new genes, over 1000 new transcripts, new miRNAs, lncRNAs, and new alternative 3'UTRs. Having high-resolution transcription data is an essential aspect of developing zebrafish as an essential model organism, and a major strength of the paper is that these data will all be easily accessed from Ensembl. Without a doubt, this work will be a valuable resource not only for zebrafish researchers, but also for developmental biologists in general.

Essential revisions:

The following issues should be addressed (in terms of analysis and discussion of the data):

1) Less complexity in terms of genes detected is observed at 50% epiboly, shield and 75% epiboly – is this of biological relevance or simply a consequence of less sequencing depth. In any case, this should be clarified and discussed.

2) Due to the experimental approach, the data reflects gene groups per stage based on the maximum expression. Hence, fine detail and developmental programmes are important but less predominant embryonic populations will be missed. This point and the limitations of the analysis should be discussed; suggesting that less represented developmental populations should be studied in isolation.

3) While observation about the excess of uncharacterised genes in gastrulation is interesting, it remains just that – an observation – and, as such, it is given too much importance in the text. The authors could do a more thorough study of uncharacterised genes, analyse types of proteins encoded by those and discuss the groups that are conserved across species or species specific; otherwise, this part should be attenuated and cut down.

4) Paralogous gene analysis shows a general trend, but is less meaningful in terms of divergence of expression patterns at a given stage, since paralogues often have different spatial distribution at the same developmental point. Hence the discussion of the shortcoming of the analysis should be added.

5) Not enough attention is attributed to the description of novel lincRNAs. Given that the two previous studies found very few overlapping candidates, the authors should discuss how many of previously identified lincRNAs they confirmed. Furthermore, with respect to the ~2000 novel lincRNAs, in addition to general levels of expression (heatmap), more information should be provided about their splicing patterns, promoter conservation metrics, etc. This is important info as the lincRNA discovery in this study nearly doubles what was previously annotated.

6) The analysis of pre-miRNA, well designed experimentally – with the whole embryo sequencing of Drosha and Dgcr8 mutants, is useful but does not naturally fit this manuscript. This part should be better integrated into the paper (e.g., the section "Novel primary miRNA transcripts" should be moved back-to-back with section "Novel features") and also put in perspective with previously identified miRNAs in zebrafish.

---

## [Author Response]

Essential revisions:The following issues should be addressed (in terms of analysis and discussion of the data):1) Less complexity in terms of genes detected is observed at 50% epiboly, shield and 75% epiboly – is this of biological relevance or simply a consequence of less sequencing depth. In any case, this should be clarified and discussed.

We believe this represents a real phenomenon, which reflects part of the maternal to zygotic transition. To check, we have downsampled each sample to 2 million read pairs per sample and recounted. The number of genes detected by the same criteria in this set is slightly smaller for all samples, but follows the same pattern of a large drop at gastrula stages.

We next examined the overlap between the genes detected in adjacent stages and this shows that at early stages there are relatively few genes that are no longer detected in the subsequent stage (column ‘lost’ in the table below) and a larger number of genes newly detected (column ‘gained’) which is why the number of detected genes goes up. At 50% epiboly, the number of genes no longer detected is higher than the number of newly detected genes, leading to an overall decrease in the number of genes. This effect is even larger at Shield stage.

**Author response image 1. respfig1:** Gene detection with reads downsampled to 2 million read pairs per sample.

**Author response image 2. respfig2:** Comparison of genes detected from one stage to the next.

**Stage1****Stage2****lost****Shared****gained**1-cell2-cell2671205113342-cell128-cell321130641851128-cell1k-cell612143037301k-cellDome756142771841Dome50%-epiboly134814770112950%-epibolyShield139314506667Shield75%-epiboly97114202115475%-epiboly1-4-somites8531450324731-4-somites14-19-somites10271594991314-19-somites20-25-somites44716415152120-25-somitesPrim-5591173451195Prim-5Prim-1558617954909Prim-15Prim-2576918094772Prim-25Long-pec526183401043Long-pecProtruding-mouth378190051608Protruding-mouthDay-432720286743Day-4Day-540320626424

Our interpretation of this is that maternally expressed genes are being cleared from the embryo by miR-430 and other pathways and the number of newly expressed zygotic genes does not compensate for this loss. We have added a sentence highlighting this in the results (subsection “A high-resolution transcriptional profile of zebrafish development”).

2) Due to the experimental approach, the data reflects gene groups per stage based on the maximum expression. Hence, fine detail and developmental programmes are important but less predominant embryonic populations will be missed. This point and the limitations of the analysis should be discussed; suggesting that less represented developmental populations should be studied in isolation.

We agree. The maximum stage groupings were used primarily to order the heatmap. As outlined below we have removed the GO and ZFA enrichment to make the manuscript more compact as it confirms what was already known rather than being genuinely novel. The BioLayout analysis provides more detail with respect to expression correlation and potential functional groups. We have added text to the discussion to acknowledge the limitations of the whole embryo approach and emphasize the need for future detailed tissue- and cell-specific studies (Discussion section).

3) While observation about the excess of uncharacterised genes in gastrulation is interesting, it remains just that – an observation – and, as such, it is given too much importance in the text. The authors could do a more thorough study of uncharacterised genes, analyse types of proteins encoded by those and discuss the groups that are conserved across species or species specific; otherwise, this part should be attenuated and cut down.

We agree and have examined the distribution of Pfam domains within these sets. The largest set of domains within the unnamed genes are zinc finger domains. These belong to the zinc finger genes in BioLayout clusters 7, 23, 26 and 135.

**Author response image 3. respfig3:** 

We retrieved Pfam domains annotated to genes from the Ensembl database and counted the number of genes for each domain. We placed domains which appear less than ten times for a stage in the Other category (yellow) and counted genes with no annotated Pfam domain as NA (grey). Three different ZnF domains were aggregated into a single ZnF category (blue). As would be expected, the named genes with vertebrate orthologues (top right) have a lot of different Pfam domains associated with them. In contrast, the unnamed genes (bottom panels) are mostly in the Other category. At gastrulation stages the largest sets are the ZnF domains across all three orthology levels.

Therefore, we have shortened the section about the spike of unnamed genes, removed it from the Discussion and now mention the overlap in the section on the ZnF genes (subsection “A burst of transcription of highly related zinc finger proteins during zygotic genome activation”).

4) Paralogous gene analysis shows a general trend, but is less meaningful in terms of divergence of expression patterns at a given stage, since paralogues often have different spatial distribution at the same developmental point. Hence the discussion of the shortcoming of the analysis should be added.

We agree and have expanded the section in the discussion on the caveats of this analysis and mention it in the Results section as well.

5) Not enough attention is attributed to the description of novel lincRNAs. Given that the two previous studies found very few overlapping candidates, the authors should discuss how many of previously identified lincRNAs they confirmed. Furthermore, with respect to the ~2000 novel lincRNAs, in addition to general levels of expression (heatmap), more information should be provided about their splicing patterns, promoter conservation metrics, etc. This is important info as the lincRNA discovery in this study nearly doubles what was previously annotated.

We would like to thank the reviewers for this suggestion. We have expanded our analysis of the lincRNAs to include data on length, exon and differential transcript isoform use and expression levels (subsection “Novel features”). We have also performed a comparative analysis with the previous two datasets (subsection “Novel features”). Furthermore, we have attempted a promoter analysis. However, in contrast to human and mouse the 5’ends of the zebrafish lincRNAs cannot be well defined currently. This is due to the lack of supporting evidence in the form of CAGE, H3K4me3 and H3K36me3 data which are used in human and mouse lincRNA calling pipeline to define the genomic promoter and gene body features. We have incorporated these findings and differences in the Results and Discussion (subsection “Novel features” and subsection “Novel Genes”).

6) The analysis of pre-miRNA, well designed experimentally – with the whole embryo sequencing of Drosha and Dgcr8 mutants, is useful but does not naturally fit this manuscript. This part should be better integrated into the paper (e.g., the section "Novel primary miRNA transcripts" should be moved back-to-back with section "Novel features") and also put in perspective with previously identified miRNAs in zebrafish.

We have moved the section and expanded the Introduction regarding the roles of miRNAs in zebrafish development. We have also added screenshots to the figures to show how to display the pri-miRNA in Ensembl (subsection “Novel primary miRNA transcripts” and subsection “Novel Genes”).